# Does Worst-Performing Agent Lead the Pack? Analyzing Agent Dynamics in Unified Distributed SGD

**Jie Hu**     **Yi-Ting Ma**     **Do Young Eun**
Department of Electrical and Computer Engineering
North Carolina State University
{jhu29, yma42, dyeun}@ncsu.edu

## Abstract

Distributed learning is essential to train machine learning algorithms across *heterogeneous* agents while maintaining data privacy. We conduct an asymptotic analysis of Unified Distributed SGD (UD-SGD), exploring a variety of communication patterns, including decentralized SGD and local SGD within Federated Learning (FL), as well as the increasing communication interval in the FL setting. In this study, we assess how different sampling strategies, such as *i.i.d.* sampling, shuffling, and Markovian sampling, affect the convergence speed of UD-SGD by considering the impact of agent dynamics on the limiting covariance matrix as described in the Central Limit Theorem (CLT). Our findings not only support existing theories on linear speedup and asymptotic network independence, but also theoretically and empirically show how efficient sampling strategies employed by individual agents contribute to overall convergence in UD-SGD. Simulations reveal that a few agents using highly efficient sampling can achieve or surpass the performance of the majority employing moderately improved strategies, providing new insights beyond traditional analyses focusing on the worst-performing agent.

## 1   Introduction

Distributed learning deals with the training of models across multiple agents over a communication network in a distributed manner, while addressing the challenges of privacy, scalability, and high-dimensional data [11, 55]. Each agent $i \in [N]$ holds a private dataset $\mathcal{X}_i$ and an agent-specified loss function $F_i : \mathbb{R}^d \times \mathcal{X}_i \to \mathbb{R}$ that depends on the model parameter $\theta \in \mathbb{R}^d$ and a data point $X \in \mathcal{X}_i$. The goal is then to find a local minima $\theta^*$ of the objective function $f(\theta) \triangleq \frac{1}{N} \sum_{i=1}^{N} f_i(\theta)$, where agent $i$'s loss function $f_i(\theta) \triangleq \mathbb{E}_{X \sim \mathcal{D}_i}[F_i(\theta, X)]$ and $\mathcal{D}_i$ represents the target distribution of data for agent $i$.[1] Each agent $i$ can locally compute the gradient $\nabla F_i(\theta, X) \in \mathbb{R}^d$ w.r.t. $\theta$ for every sampled data point $X \in \mathcal{X}_i$. Due to the distributed nature, $\{\mathcal{D}_i\}_{i \in [N]}$ and $\{\mathcal{X}_i\}_{i \in [N]}$ are not necessarily identically distributed over $[N]$ so that the minima of each local function $f_i(\theta)$ can be far away from $\mathcal{L}$. This is particularly relevant in decentralized training data, e.g., Federated Learning (FL) with *heterogeneous* data across data centers or devices [81, 31].

In this paper, we focus on Unified Distributed SGD (UD-SGD), where each agent $i \in [N]$ updates its model parameter $\theta_{n+1}^i$ in a two-step process:

$$\text{\textit{Local update:} } \theta_{n+1/2}^i = \theta_n^i - \gamma_{n+1} \nabla F_i(\theta_n^i, X_n^i), \tag{1a}$$

---

[1] Throughout the paper we don't impose convexity assumption on $f(\theta)$. For convex $f(\theta)$, $\mathcal{L}$ is the global minima. For non-convex $f(\theta)$, $\mathcal{L}$ represents the collection of local minima, which is of great interest in neural network training for sufficiently good performance [20, 19]. With an additional condition such as the Polyak-Lojasiewicz inequality, non-convex $f(\theta)$ is ensured to have a unique minima [1, 75, 78].

38th Conference on Neural Information Processing Systems (NeurIPS 2024).

$$\text{Aggregation: } \theta_{n+1}^i = \sum_{j=1}^N w_n(i,j)\theta_{n+1/2}^j, \tag{1b}$$

where $\gamma_n$ denotes the step size, $X_n^i$ is the data sampled by agent $i$ at time $n$ (i.e., agent dynamics), and $\mathbf{W}_n = [w_n(i,j)]_{i,j\in[N]}$ represents the doubly-stochastic communication matrix satisfying $w_n(i,j) \geq 0$ and $\mathbf{1}^T\mathbf{W}_n = \mathbf{1}^T$, $\mathbf{W}_n\mathbf{1} = \mathbf{1}$. In the special case of $N = 1$, (1) simplifies to the vanilla SGD where $\mathbf{W}_n = 1$ for all $n$. UD-SGD covers a wide range of distributed algorithms, e.g., decentralized SGD (DSGD) [71, 80, 61, 68], distributed SGD with changing topology (DSGD-CT) [24, 43], local SGD (LSGD) in FL [55, 76], and its variant aimed at reducing communication costs (LSGD-RC) [51].

**Versatile Communication Patterns $\{\mathbf{W}_n\}$:**
For visualization, we depict the scenarios of UD-SGD (1) in Figure 1. In DSGD, each agent (node) in the graph communicates with its neighbors after each SGD computation via $\mathbf{W}_n$, representing the underlying network topology. As a special case, central server-based aggregation, forming a fully connected network, translates $\mathbf{W}_n$ into a rank-1 matrix $\mathbf{W}_n = \mathbf{11}^T/N$. To minimize communication expenses, FL variants allow each agent to perform multiple SGD steps before aggregation [55, 67, 76], resulting in a communication interval of length $K$ and a consistent pattern $\mathbf{W}_n = \mathbf{W}$ for $n = mK, \forall m \in \mathbb{N}$, and $\mathbf{W}_n = \mathbf{I}_N$ otherwise. In particular, i)

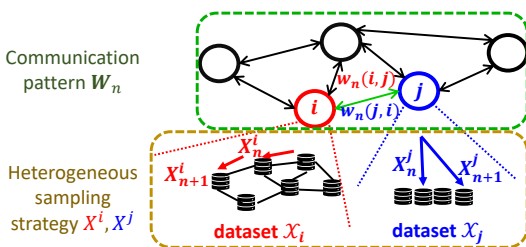

Figure 1: GD-SGD algorithm with a communication network of $N = 5$ agents, each holding potentially distinct datasets; e.g., agent $j$ (in blue) samples $\mathcal{X}_j$ *i.i.d.* and agent $i$ (in red) samples $\mathcal{X}_i$ via Markovian trajectory.

$\mathbf{W} = \mathbf{11}^T/N$ corresponds to LSGD with full agent participation (LSGD-FP) [76, 42, 51]; ii) $\mathbf{W}$ is a random matrix generated by partial agent participation (LSGD-PP) [55, 18, 74]; iii) $\mathbf{W}$ is generated by Metropolis-Hasting algorithm in decentralized setting, e.g., hybrid LSGD (HLSGD) [37, 32] and decentralized FL (DFL) [46, 77, 16]. We defer further discussion of $\mathbf{W}$ to Appendix F.1.

**Markovian vs *i.i.d.* Sampling:** Agents typically employ *i.i.d.* or Markovian sampling, as illustrated in the bottom brown box of Figure 1. In cases where agents have full access to their data, DSGD with *i.i.d* sampling has been extensively studied [60, 43, 61, 47]. In FL, many application-oriented LSGD variants have been investigated [51, 18, 77, 32, 37, 53]. However, these works solely focus on *i.i.d.* sampling, restricting their applicability to Markovian sampling scenarios.

Markovian sampling, which has received increased attention in limited settings (see Table 1), is vital where agents lack independent data access. For instance, in statistical applications, agents with an unknown a priori distribution often use Markovian sampling over *i.i.d.* sampling [40, 63]. In HLSGD across device-to-device (D2D) networks [32, 37], random walks reduce communication costs compared to the frequent aggregations required by Gossip algorithms [38, 28, 4]. For single-agent scenarios, vanilla SGD with Markovian noise, as applied in a D2D network, has shown improved communication efficiency and privacy [69, 28, 35]. In contrast, for agents with full data access, Markov Chain Monte Carlo (MCMC) methods can be more efficient than *i.i.d.* sampling, especially in high-dimensional spaces with constraints [27, 40], where acceptance-rejection methods [12] lead to computational inefficiency (e.g., wasted samples) due to multiple rejections before obtaining a sample that satisfies constraints [26, 69]. In addition, shuffling methods can be considered as high-order Markov chains [38], which achieves faster convergence than *i.i.d.* sampling [1, 79, 78].

**Limitations of Non-Asymptotic Analysis on Agent's Sampling Strategy:** Recent studies on the non-asymptotic behavior of DSGD and LSGD variants under Markovian sampling, as summarized in Table 1, have made significant strides. However, these works often fall short in accurately revealing the statistical influence of *each* agent dynamics $\{X_n^i\}$ on the performance of UD-SGD. For instance, [71, 68] proposed the error bound $O(\frac{1/\log^2(1/\rho)}{n^{1-a}})$, where $a \in (0.5, 1]$ and $\rho$ denotes the identical mixing rate for all agents, overlooking agent heterogeneity in sampling strategy. A similar assumption to $\rho$ is also evident in [42]. More recent contributions from [80, 72] have attempted to relax these constraints by considering a finite-time bound of $O(\tau_{mix}^2/(n+1))$, where $\tau_{mix}$ is the mixing time of the slowest agent. This approach, however, inherently focuses on *the worst-performing agent*, neglecting how other agents with faster mixing rates might positively influence the system.[2] Such an analysis fails to capture the collective impact of *other* agents on the overall system performance,

---

[2]Although improving the finite-time upper bound to distinguish each agent may not be the focus of the aforementioned works, their analyses require every Markov chain to be close to some neighborhood of its

Table 1: Comparison of recent works in distributed learning: We classify the communication patterns into seven categories, i.e., DSGD, DSGD-CT, LSGD-FP, LSGD-PP, LSGD-RC, HLSGD and DFL. We mark 'UD-SGD' when all aforementioned patterns are included and the detailed discussion on $\{\mathbf{W}_n\}$ is referred to Appendix F.1. Abbreviations: '**Asym.**' = 'Asymptotic', '**D.A.B**' = 'Differentiating Agent Behavior', '**L.S.**' = 'Linear Speedup', '**A.N.I.**' = 'Asymptotic Network Independence'.

| Reference | Analysis | Sampling | Communication Pattern | D.A.B. | L.S. | A.N.I. |
|-----------|----------|----------|-----------------------|--------|------|--------|
| [58] | Asym. | *i.i.d.* | DSGD | ✓ | ✓ | ✓ |
| [51] | Asym. | *i.i.d.* | LSGD-RC | ✓ | ✓ | N/A |
| [43, 47] | Non-Asym. | *i.i.d.* | DSGD-CT | ✗ | ✓ | ✗ |
| [61] | Non-Asym. | *i.i.d.* | DSGD | ✗ | ✗ | ✓ |
| [18, 53] | Non-Asym. | *i.i.d.* | LSGD-PP | ✗ | ✓ | N/A |
| [37, 32] | Non-Asym. | *i.i.d.* | HLSGD | ✗ | ✓ | ✗ |
| [77, 16] | Non-Asym. | *i.i.d.* | DFL | ✗ | ✓ | ✗ |
| [71, 80, 68] | Non-Asym. | Markov | DSGD | ✗ | ✗ | ✗ |
| [42, 72] | Non-Asym. | Markov | LSGD-FP | ✗ | ✓ | N/A |
| [69, 4, 28] | Non-Asym. | Markov | N/A (single agent) | N/A | N/A | N/A |
| [38, 52] | Asym. | Markov | N/A (single agent) | N/A | N/A | N/A |
| Our Work | Asym. | Markov | UD-SGD | ✓ | ✓ | ✓ |

a crucial aspect in large-scale applications where identifying and managing the worst-performing agent is challenging due to privacy concerns or sporadic unreachability. Since agents in distributed learning have the freedom to choose their sampling strategies, it's vital to understand how each agent's improved sampling approach contributes to the overall convergence speed of the UD-SGD algorithm. This understanding is key to enhancing system performance, particularly in large-scale machine learning scenarios where agent heterogeneity is a defining feature.

**Rationale for Asymptotic Analysis:** Recent trends in convergence analysis have leaned towards non-asymptotic methods, yet it's crucial to recognize the complementary role of asymptotic analysis for a better understanding of convergence behaviors, as highlighted in [9, 56, 25, 39]. For vanilla SGD, [59, 17] emphasized that central limit theorem (CLT) is far less asymptotic than it may appear under both *i.i.d.* and Markovian sampling. Notably, the limiting covariance matrix, a key statistical feature in vanilla SGD's CLT, also prominently features in high-probability bound [59], explicit finite-time bound [17] and 1-Wasserstein distance in the non-asymptotic CLT [66]. [38] further underscored this by numerically showing that the limiting covariance matrix provides a more precise depiction of convergence than the mixing rates often used in finite-time upper bounds [26, 69]. Moreover, they argued that finite-time analysis may not suitably apply to certain efficient high-order Markov chains, due to the lack of comparative mixing-rate metrics.

**Our Contributions:** We present an asymptotic analysis of the UD-SGD algorithm (1) under heterogeneous agent dynamics $\{X_n^i\}$ and a large family of communication patterns $\{\mathbf{W}_n\}$. Specifically,

• Under appropriate assumptions, all agents performing (1) asymptotically reach the consensus and find $\theta^*$: $\forall i \in [N]$, $\theta_n \triangleq \frac{1}{N}\sum_{i=1}^N \theta_n^i$ denotes the average model parameter among all agents, we have

$$\lim_{n\to\infty}\|\theta_n^i - \theta_n\| = 0, \quad \lim_{n\to\infty}\|\theta_n - \theta^*\| = 0 \text{ a.s.} \tag{2}$$

Moreover, we derive the CLT of UD-SGD in the form of

$$\gamma_n^{-1/2}(\theta_n - \theta^*) \xrightarrow[n\to\infty]{dist.} \mathcal{N}(\mathbf{0}, \mathbf{V}). \tag{3}$$

Our framework addresses technical challenges in quantifying consensus error under various communication patterns and slowly increasing communication interval. This shows a substantial extension compared to previous studies [58, 43, 51], particularly in regulating the growth of communication

---

stationary distribution. This naturally incurs a *maximum operator*, and thus convergence is strongly influenced by the *slowest* mixing rate, i.e., the worst-performing agent.

intervals (Assumption 2.3-ii) and in proving the scaled consensus error's boundedness (Lemma B.1). Furthermore, we reformulate UD-SGD as a stochastic approximation-like iteration and tackle the Markovian noise term using the Poisson equation, a technique previously confined only to vanilla SGD with Markovian sampling [17, 38, 52]. The key here is to devise the noise decomposition that separates the consensus error among all agents from the error caused by the bias from the Markov chain, which aligns with the target distribution only asymptotically at infinity, not at finite times.

• In analyzing (3), we derive the exact form of $\mathbf{V}$ as $\frac{1}{N^2} \sum_{i=1}^{N} \mathbf{V}_i$. Here, $\mathbf{V}_i$ is the limiting covariance matrix of agent $i$, which depends mainly on its sampling strategy $\{X_n^i\}$. This allows us to show that improving *individual* agents' sampling strategy can reduce the covariance in CLT, which in turn implies a smaller mean-square error (MSE) for large time $n$. This is a significant advancement over previous finite-sample bounds that only account for the worst-performing agent and do not fully capture the effect of individual agent dynamics on overall system performance. Our CLT result (3) also treats recent findings in [38] as a very special case with $N = 1$, where the relationship therein between the sampling efficiency of the Markov chain and the limiting covariance matrix in the CLT of vanilla SGD, can carry over to our UD-SGD.

• We demonstrate that our analysis supports recent findings from studies such as [42], which exhibited linear speedup scaling with the number of agents under LSGD-FP with Markovian sampling; and [62, 61], which examined the notion of 'asymptotic network independence' for DSGD with *i.i.d.* sampling, where the convergence of the algorithm (1) at large time $n$ depends solely on the left eigenvector of $\mathbf{W}_n$ ($\frac{1}{N}\mathbf{1}$ considered in this paper) rather than the specific communication network topology encoded in $\mathbf{W}_n$, but now under Markovian sampling. We extend these findings in view of CLT to a broader range of communication patterns $\{\mathbf{W}_n\}$ and general sampling strategies $\{X_n^i\}$.

• We conduct numerical experiments using logistic regression and neural network training with several choices of agents' sampling strategies, including a recently proposed one via nonlinear Markov chain [25]. Our results uncover a key phenomenon: a handful of compliant agents adopting highly efficient sampling strategies can match or exceed the performance of the majority using moderately improved strategies. This finding is crucial for practical optimization in large-scale learning systems, moving beyond the current literature that only considers the worst-performing agent in more restrictive settings.

## 2 Preliminaries

**Basic Notations:** We use $\|\mathbf{v}\|$ to indicate the Euclidean norm of a vector $\mathbf{v} \in \mathbb{R}^d$ and $\|\mathbf{M}\|$ to indicate the spectral norm of a matrix $\mathbf{M} \in \mathbb{R}^{d \times d}$. The identity matrix of dimension $d$ is denoted by $\mathbf{I}_d$, and the all-one (resp. all-zero) vector of dimension $N$ is denoted by $\mathbf{1}$ (resp. $\mathbf{0}$). Let $\mathbf{J} \triangleq \mathbf{1}\mathbf{1}^T/N$. The diagonal matrix with the entries of $\mathbf{v}$ on the main diagonal is written as diag($\mathbf{v}$). We also use '$\succeq$' for Loewner ordering such that $\mathbf{A} \succeq \mathbf{B}$ is equivalent to $\mathbf{x}^T(\mathbf{A} - \mathbf{B})\mathbf{x} \geq 0$ for any $\mathbf{x} \in \mathbb{R}^d$.

**Asymptotic Covariance Matrix:** Asymptotic variance is a widely used metric for evaluating the second-order properties of Markov chains associated with a scalar-valued test function in the MCMC literature, e.g., Chapter 6.3 [12], and asymptotic covariance matrix is its multivariate version for a vector-valued function. Specifically, we consider a finite, irreducible, aperiodic and positive recurrent (ergodic) Markov chain $\{X_n\}_{n \geq 0}$ with transition matrix $\mathbf{P}$ and stationary distribution $\boldsymbol{\pi}$, and the estimator $\hat{\mu}_n(\mathbf{g}) \triangleq \frac{1}{n}\sum_{s=0}^{n-1} \mathbf{g}(X_s)$ for any vector-valued function $\mathbf{g} : [N] \rightarrow \mathbb{R}^d$. According to the ergodic theorem [12, 13], we have $\lim_{n \to \infty} \hat{\mu}_n(\mathbf{g}) = \mathbb{E}_{\boldsymbol{\pi}}(\mathbf{g})$ a.s.. As defined in [13, 38], the asymptotic covariance matrix $\boldsymbol{\Sigma}_X(\mathbf{g})$ for a vector-valued function $\mathbf{g}(\cdot)$ is given by

$$\boldsymbol{\Sigma}_X(\mathbf{g}) \triangleq \lim_{n \to \infty} n \cdot \text{Var}(\hat{\mu}_n(\mathbf{g})) = \lim_{n \to \infty} \frac{1}{n} \cdot \mathbb{E}\left\{\Delta_n \Delta_n^T\right\}, \tag{4}$$

where $\Delta_n \triangleq \sum_{s=0}^{n-1}(\mathbf{g}(X_s) - \mathbb{E}_{\boldsymbol{\pi}}(\mathbf{g}))$. By following the algebraic manipulations in [12, Theorem 6.3.7] for asymptotic variance (univariate version), we can rewrite (4) in a matrix form such that

$$\boldsymbol{\Sigma}_X(\mathbf{g}) = \mathbf{G}^T \text{diag}(\boldsymbol{\pi}) \left(\mathbf{Z} - \mathbf{I}_N + \mathbf{1}\boldsymbol{\pi}^T\right) \mathbf{G}, \tag{5}$$

where $\mathbf{G} \triangleq [\mathbf{g}(1), \cdots, \mathbf{g}(N)]^T \in \mathbb{R}^{N \times d}$ and $\mathbf{Z} \triangleq [\mathbf{I}_N - \mathbf{P} + \mathbf{1}\boldsymbol{\pi}^T]^{-1}$. This matrix form explicitly shows the dependence on the transition matrix $\mathbf{P}$ and its stationary distribution $\boldsymbol{\pi}$, and will be utilized in our Theorem 3.3.

**Model Description:** The UD-SGD in (1) can be expressed in a compact iterative form, i.e., we have

$$\theta_{n+1}^i = \sum_{j=1}^N w_n(i,j)(\theta_n^j - \gamma_{n+1}\nabla F_j(\theta_n^j, X_n^j)), \tag{6}$$

at each time $n$, where each agent $i$ samples according to its own Markovian trajectory $\{X_n^i\}_{n\geq 0}$ with stationary distribution $\pi_i$ such that $\mathbb{E}_{X\sim\pi_i}[F_i(\theta, X)] = f_i(\theta)$. Let $K_l$ denote the communication interval between the $(l-1)$-th and $l$-th aggregation among $N$ agents, and $n_l \triangleq \sum_{m=1}^l K_m$ be the time instance for the $l$-th aggregation. We also define $\tau_n \triangleq \min_l\{l : n_l \geq n\}$ as the index of the upcoming aggregation at time $n$ such that $K_{\tau_n}$ indicates the communication interval for the $\tau_n$-th aggregation, or more precisely, the length of the communication interval that includes the time index $n$. The communication pattern follows that $\mathbf{W}_n = \mathbf{I}_n$ if $n \neq n_l$ and $\mathbf{W}_n = \mathbf{W}$ otherwise for $l \geq 1$, where the examples of $\mathbf{W}$ will be discussed in Appendix F.1. Note that i) when $K_l = 1$, (6) reduces to DSGD; ii) when $K_l = K > 1$, (6) becomes the local SGD in FL. iii) When $K_l$ increases with $l$, we recover some choices of $K_l$ studied in [51] beyond LSGD-RC with *i.i.d.* sampling. This increasing communication interval aims to further reduce the frequency of aggregation among agents for lower communication costs, but now under a Markovian sampling setting and a wider range of communication patterns. We below state the assumptions needed for the main theoretical results.

**Assumption 2.1** (Regularity of the gradient). *For each $i \in [N]$ and $X \in \mathcal{X}^i$, the function $F_i(\theta, X)$ is L-smooth in terms of $\theta$, i.e., for any $\theta_1, \theta_2 \in \mathbb{R}^d$,*

$$\|\nabla F_i(\theta_1, X) - \nabla F_i(\theta_2, X)\| \leq L\|\theta_1 - \theta_2\|. \tag{7}$$

*In addition, we assume that the objective function $f$ is twice continuously differentiable and $\mu$-strongly convex only around the local minima $\theta^* \in \mathcal{L}$, i.e.,*

$$\mathbf{H} \triangleq \nabla^2 f(\theta^*) \succeq \mu \mathbf{I}_d. \tag{8}$$

Assumption 2.1 imposes the regularity conditions on the gradient $\nabla F_i(\cdot, X)$ and Hessian matrix of the objective function $f(\cdot)$, as is commonly assumed in [10, 45, 29, 38]. Note that (7) requires per-sample Lipschitzness of $\nabla F_i$ and is stronger than the Lipschitzness of its expected version $\nabla f_i$, which is commonly assumed under *i.i.d* sampling setting [73, 50, 30]. However, we remark that this is in line with previous work on DSGD and LSGD-FP under Markovian sampling as well [71, 42, 80], because $\nabla F_i(\theta, X)$ is no longer the unbiased stochastic version of $\nabla f_i(\theta)$ and the effect of $\{X_n^i\}$ has to be taken into account in the analysis. The local strong convexity at the minimizer is commonly assumed to analyze the convergence of the algorithm under both asymptotic and non-asymptotic analysis [10, 29, 38, 45, 52, 80].

**Assumption 2.2** (Ergodicity of Markovian sampling). *$\{X_n^i\}_{n\geq 0}$ is an ergodic Markov chain with stationary distribution $\boldsymbol{\pi}_i$ such that $\mathbb{E}_{X\sim\boldsymbol{\pi}_i}[F_i(\theta, X)] = f_i(\theta)$, and is independent from $\{X_n^j\}_{n\geq 0}, j \neq i$.*

The ergodicity of the underlying Markov chains, as stated in Assumption 2.2, is commonly assumed in the literature [26, 69, 80, 42, 38]. This assumption ensures the asymptotic unbiasedness of the loss function $F_i(\theta, \cdot)$, which takes *i.i.d.* sampling as a special case.

**Assumption 2.3** (Decreasing step size and slowly increasing communication interval). *i) For bounded communication interval $K_{\tau_n} \leq K, \forall n$, we assume the polynomial step size $\gamma_n = 1/n^a$ and $a \in (0.5, 1]$; Or ii) If $K_{\tau_n} \to \infty$ as $n \to \infty$, we assume $\gamma_n = 1/n$ and define $\eta_n = \gamma_n K_{\tau_n}^{L+1}$, where the sequence $\{K_l\}_{l\geq 0}$ satisfies $\sum_n \eta_n^2 < \infty$, $K_{\tau_n} = o(\gamma_n^{-1/2(L+1)})$, and $\lim_{l\to\infty}\eta_{n_l+1}/\eta_{n_{l+1}+1} = 1$.*

In Assumption 2.3, the polynomial step size $\gamma_n$ is standard in the literature and it has the property $\sum_n \gamma_n = \infty$, $\sum_n \gamma_n^2 < \infty$ [17, 38]. Inspired by [51], we introduce $\eta_n$ to control the step size within each $l$-th communication interval with length $K_l$ to restrict the growth of $K_l$. Specifically, $\sum_n \eta_n^2 < \infty$ and $K_{\tau_n} = o(\gamma_n^{-1/2(L+1)})$ ensure that $\eta_n \to 0$ and $K_{\tau_n}$ does not increase too fast in $n$. $\lim_{l\to\infty}\eta_{n_l+1}/\eta_{n_{l+1}+1} = 1$ sets the restriction on the increment from $n_l$ to $n_{l+1}$. Several practical forms of $K_l$ suggested by [51], including $K_l \sim \log(l)$ and $K_l \sim \log\log(l)$, also satisfy Assumption 2.3-ii). We defer to Appendix A the mathematical verification of these two types of $K_l$, together with the practical implications of increasing communication interval $K_l$.

**Remark 1.** *In Assumption 2.3, we incorporate an increasing communication interval along with a step size $\gamma_n = 1/n$. This complements the choice of step size $\gamma_n$ in [51, Assumption 3.3], where $\gamma_n = 1/n^a$ for $a \in (0.5, 1)$. It is important to note, however, that the increasing communication interval specified in [51, Assumption 3.2] is applicable only in i.i.d sampling. Under the Markovian*

*sampling framework, the expression $\nabla F_i(\theta, X) - \nabla f_i(\theta)$ loses its unbiased and Martingale difference properties. Consequently, the Martingale CLT application as utilized by [51] does not directly extend to Markovian sampling. To address this, we adapted techniques from [29, 58] to accommodate the increasing communication interval within the Markovian sampling setting and various communication patterns. This adaptation necessitates $\gamma_n = 1/n$, a specification not covered in [51]. Exploring more general forms of $K_l$ that could relax this assumption is outside the scope of our current study.*

**Assumption 2.4** (Stability on model parameter). *We assume $\sup_n \|\theta_n^i\| < \infty$ almost surely $\forall i \in [N]$.*

Assumption 2.4 claims that the sequence of $\{\theta_n^i\}$ always remains in a path-dependent compact set. It is to ensure the stability of the algorithm that serves the purpose of analyzing the convergence, which is often assumed under the asymptotic analysis of vanilla SGD with Markovian noise [23, 29, 52]. As mentioned in [58, 70], checking Assumption 2.4 is challenging and requires case-by-case analysis, even under *i.i.d.* sampling. Only recently the stability of SGD under Markovian sampling has been studied in [9], but the result for UD-SGD remains unknown in the literature. Thus, we analyze each agent's sampling strategy in the asymptotic regime under this stability condition.

**Assumption 2.5** (Contraction property of communication matrix). *i). $\{\mathbf{W}_n\}_{n \geq 0}$ is independent of the sampling strategy $\{X_n^i\}_{n \geq 0}$ for all $i \in [N]$ and is assumed to be doubly-stochastic for all $n$; ii). At each aggregation step $n_l$, $\mathbf{W}_{n_l}$ is independently generated from some distribution $\mathcal{P}_{n_l}$ such that $\|\mathbb{E}_{\mathbf{W} \sim \mathcal{P}_{n_l}}[\mathbf{W}^T \mathbf{W}] - \mathbf{J}\| \leq C_1 < 1$ for some constant $C_1$.*

The doubly-stochasticity of $\mathbf{W}_n$ in Assumption 2.5-i) is widely assumed in the literature [54, 24, 43, 80]. Assumption 2.5-ii) is a contraction property to ensure that agents employing UD-SGD will asymptotically achieve the consensus, which is also common in [7, 24, 80]. Examples of $\mathbf{W}$ that satisfy Assumption 2.5-ii), e.g., Metropolis-Hasting matrix, partial agent participation in FL, are deferred to Appendix F.1 due to space constraint.

## 3 Asymptotic Analysis of UD-SGD

**Almost Sure Convergence:** Let $\theta_n \triangleq \frac{1}{N} \sum_{i=1}^N \theta_n^i$ represent the consensus among all the agents at time $n$, we establish the asymptotic consensus of the local parameters $\theta_n^i$, as stated in Lemma 3.1.

**Lemma 3.1.** *Under Assumptions 2.1, 2.3, 2.4 and 2.5, the consensus error $\theta_n^i - \theta_n$ diminishes to zero at the rate specified below: Almost surely, for every agent $i \in [N]$,*

$$\|\theta_n^i - \theta_n\| = \begin{cases} O(\gamma_n) & \text{under Assum. 2.3-i),} \\ O(\eta_n) & \text{under Assum. 2.3-ii).} \end{cases} \tag{9}$$

Lemma 3.1 indicates that all agents asymptotically reach consensus at a rate of $O(\gamma_n)$ (or $O(\eta_n)$). This finding extends the scope of [58, Proposition 1], incorporating considerations for Markovian sampling, FL settings, and increasing communication interval $K_l$. The proof, detailed in Appendix B, primarily tackles the challenge of establishing the boundedness of the sequences $\{\gamma_n^{-1}(\theta_n^i - \theta_n)\}$ (or $\{\eta_n^{-1}(\theta_n^i - \theta_n)\}$) almost surely for all $i \in [N]$. This is specifically analyzed in Lemma B.1. Next, with additional Assumption 2.2, we are able to obtain the almost sure convergence of $\theta_n$ to $\theta^* \in \mathcal{L}$.

**Theorem 3.2.** *Under Assumptions 2.1 - 2.5, the consensus $\theta_n$ converges to $\mathcal{L}$ almost surely, i.e.,*

$$\limsup_n \inf_{\theta^* \in \mathcal{L}} \|\theta_n - \theta^*\| = 0 \quad \text{a.s.} \tag{10}$$

Theorem 3.2 is achieved by decomposing the Markovian noise term $\nabla F_i(\theta_n^i, X_n^i) - \nabla f_i(\theta_n^i)$, using the Poisson equation technique as discussed in [6, 29, 17], into a Martingale difference noise term, along with additional noise terms. We then reformulate (6) into an iteration akin to stochastic approximation, as depicted in (56). The subsequent step involves verifying the conditions on these noise terms under our stated assumptions. Crucially, this theorem also establishes that UD-SGD ensures an almost sure convergence of each agent to a local minimum $\theta^* \in \mathcal{L}$, even in scenarios where the communication interval $K_l$ gradually increases, in accordance with Assumption 2.3-ii). The detailed proof of this theorem is provided in Appendix C.

**Central Limit Theorem:** Let $\mathbf{U}_i \triangleq \Sigma_{X^i}(\nabla F_i(\theta^*, \cdot))$ represent the asymptotic covariance matrix (defined in (5)) associated with each agent $i \in [N]$, given their sampling strategy $\{X_n^i\}$ and function $\nabla F_i(\theta^*, \cdot)$. Define $\mathbf{U} \triangleq \frac{1}{N^2} \sum_{i=1}^N \mathbf{U}_i$. We assume the polynomial step-size $\gamma_n \sim \gamma_\star / n^a$, $a \in (0.5, 1]$ and $\gamma_\star > 0$. In the case of $a = 1$, we further assume $\gamma_\star > 1/2\mu$, where $\mu$ is defined in (8). For notational simplicity, and without loss of generality, our remaining CLT result is stated while conditioning on the event that $\{\theta_n \to \theta^*\}$ for some $\theta^* \in \mathcal{L}$.

**Theorem 3.3.** *Let Assumptions 2.1 - 2.5 hold. Then,*

$$\gamma_n^{-1/2}(\theta_n - \theta^*) \xrightarrow[n \to \infty]{dist.} \mathcal{N}(\mathbf{0}, \mathbf{V}), \tag{11}$$

*where the limiting covariance matrix $\mathbf{V}$ is in the form of*

$$\mathbf{V} = \int_0^\infty e^{\mathbf{M}t} \mathbf{U} e^{\mathbf{M}t} dt. \tag{12}$$

*Here, we have $\mathbf{M} = -\mathbf{H}$ if $a \in (0.5, 1)$, or $\mathbf{M} = \mathbf{I}_d/2\gamma_\star - \mathbf{H}$ if $a = 1$, where $\mathbf{H}$ is defined in (8).*

*Moreover, let $\bar{\theta}_n = \frac{1}{n}\sum_{s=0}^{n-1}\theta_s$ and $\mathbf{V}' = \mathbf{H}^{-1}\mathbf{U}\mathbf{H}^{-1}$. For $a \in (0.5, 1)$, we have*

$$\sqrt{n}(\bar{\theta}_n - \theta^*) \xrightarrow[n \to \infty]{dist.} \mathcal{N}(\mathbf{0}, \mathbf{V}'), \tag{13}$$

The proof, presented in Appendix D, addresses the technical challenges in deriving the CLT for UD-SGD, specifically the second-order conditions in decomposing the Markovian noise term, which is not present in the *i.i.d.* sampling case [58, 43, 51]. We decompose $\nabla F_i(\theta_n, X_n^i) - \nabla f_i(\theta_n)$ into three parts in (48) using Poisson equation: $e_{n+1}^i, \nu_{n+1}^i, \xi_{n+1}^i$. The consensus error $\theta_n^i - \theta_n$ embedded in noise terms $e_{n+1}^i$ and $\xi_{n+1}^i$ is a new factor, whose characteristics have been quantified in our Lemma 3.1 but are not present in the single-agent scenario analyzed as an application of stochastic approximation in [22, 29]. The specifics of this analysis are expanded upon in Appendices D.1 to D.3. We require $\gamma_\star > 1/2\mu$ for $a = 1$ to ensure that the largest eigenvalue of $\mathbf{M}$ is negative, as this is a necessary condition for the existence of $\mathbf{V}$ in (12) (otherwise integration diverges). In the case where there is only one agent ($N = 1$), $\mathbf{V}$ and $\mathbf{V}'$ reduce to the matrices specified in the CLT result of vanilla SGD [29, 38, 52]. In addition, for a special case of constant communication interval in Assumption 2.3-i) and *i.i.d.* sampling as shown in Table 1, we recover the CLT of LSGD-RC in [51]. See Appendix E for detailed discussions.

Theorem 3.3 has significant implications for the MSE of $\{\theta_n\}$ for large time $n$, i.e., $\mathbb{E}[\|\theta_n - \theta^*\|^2] = \sum_{i=1}^d \mathbf{e}_i^T \mathbb{E}[(\theta_n - \theta^*)(\theta_n - \theta^*)^T]\mathbf{e}_i \approx \gamma_n \sum_{i=1}^d \mathbf{e}_i^T \mathbf{V}\mathbf{e}_i = \gamma_n \mathrm{Tr}(\mathbf{V})$, where $\mathbf{e}_i$ is the $d$-dimensional vector of all zeros except 1 at the $i$-th entry. This indicates that a smaller limiting covariance matrix $\mathbf{V}$, according to the Loewner order, results in a smaller trace of $\mathbf{V}$ and consequently in a reduced MSE for large $n$. Consideration for smaller $\mathbf{V}$ will be presented in the next section, where agents have the opportunity to improve their sampling strategies.

**Remark 2.** *Studies by [62, 61] have shown that in DSGD with a fixed doubly-stochastic matrix $\mathbf{W}$, the influence of communication topology diminishes after a transient period. Our Theorem 3.3 extends these findings to Markovian sampling and a broader spectrum of communication patterns as in Table 1. This extension is based on the fact that the consensus error, impacted primarily by the communication pattern, decreases faster than the CLT scale $O(\sqrt{\gamma_n})$ and is thus not the dominant factor in the asymptotic regime, as suggested by Lemma 3.1.*

**Remark 3.** *Recent studies have highlighted linear speedup with increasing number of agents $N$ in the dominant term of their finite-sample error bounds under DSGD-CT with i.i.d. sampling [43] and LSGD-FC with Markovian sampling [42]. However, our Theorem 3.3 demonstrates this phenomenon under more diverse communication patterns and Markovian sampling in Table 1 via the leading term $\mathbf{V}$ in our CLT. Specifically, it scales with $1/N$, i.e. $\mathbf{V} = \bar{\mathbf{V}}/N$, where $\bar{\mathbf{V}} = \frac{1}{N}\sum_{i=1}^N \mathbf{V}_i$ denotes the average limiting covariance matrices across all $N$ agents and $\mathbf{V}_i = \int_0^\infty e^{\mathbf{M}t}\mathbf{U}_i e^{\mathbf{M}t}dt$, suggesting that the MSE $\mathbb{E}[\|\theta_n - \theta^*\|^2]$ will be improved by $1/N$. A similar argument also applies to $\mathbf{V}'$ in (13), i.e., $\mathbf{V}' = \bar{\mathbf{V}}'/N$, where $\bar{\mathbf{V}}' = \frac{1}{N}\sum_{i=1}^N \mathbf{V}_i'$ and $\mathbf{V}_i' = \mathbf{H}^{-1}\mathbf{U}_i\mathbf{H}^{-1}$.*

**Impact of Agent's Sampling Strategy:** In the literature, the mixing time-based technique has been widely used in the non-asymptotic analysis in SGD, DSGD and various LSGD variants in FL [26, 69, 68, 80, 42], i.e., for each agent $i \in [N]$ and some constant $C$,

$$\|\nabla F_i(\theta, X_n^i) - \nabla f_i(\theta)\| \le C\|\theta\|\rho_i^n, \tag{14}$$

where $\rho_i$ is the mixing rate of the underlying Markov chain. However, typical non-asymptotic analyses often rely on $\rho \triangleq \max_i \rho_i$ among $N$ agents, i.e., the worst-performing agent in their finite-time bounds [80, 72], or assume an identical mixing rate across all $N$ agents [42, 68].

In contrast, Remark 3 highlights that each agent holds its own limiting covariance matrices $\mathbf{V}_i$ and $\mathbf{V}_i'$, which are predominantly governed by the matrix $\mathbf{U}_i$, capturing the agent's sampling strategy

$\{X_n^i\}$ and *contributing equally* to the overall performance of UD-SGD. For each agent $i$, denote by $\mathbf{U}_i^X$ and $\mathbf{U}_i^Y$ the asymptotic covariance matrices associated with two candidate sampling strategies $\{X_n^i\}$ and $\{Y_n^i\}$, respectively. Let $\mathbf{V}^X$ and $\mathbf{V}^Y$ be the limiting covariance matrices of the distributed system in (12), where agent $i$ employs $\{X_n^i\}$ and $\{Y_n^i\}$, respectively, while keeping other agents' sampling strategies unchanged. Then, we have the following result.

**Corollary 3.4.** *For agent $i$, if there exist two sampling strategies $\{X_n^i\}_{n\geq 0}$ and $\{Y_n^i\}_{n\geq 0}$ such that $\mathbf{U}_i^X \succeq \mathbf{U}_i^Y$, we have $\mathbf{V}^X \succeq \mathbf{V}^Y$.*

Corollary 3.4 directly follows from the definition of Loewner ordering, and Loewner ordering being closed under addition (i.e., $\mathbf{A} \succeq \mathbf{B}$ implies $\mathbf{A}+\mathbf{C} \succeq \mathbf{B}+\mathbf{C}$). It demonstrates that even a single agent improves its sampling strategy from $\{X_n^i\}$ to $\{Y_n^i\}$, it leads to an overall reduction in $\mathbf{V}$ (in terms of Loewner ordering), thereby decreasing the MSE and benefiting the entire group of $N$ agents. The subsequent question arises: *How do we identify an improved sampling strategy $\{Y_n^i\}$ over the baseline $\{X_n^i\}$?*

This question has been partially addressed by [57, 48, 38], which qualitatively investigates the 'efficiency ordering' of two sampling strategies. In particular, [38, Theorem 3.6 (i)] shows that sampling strategy $\{Y_n\}$ is more efficient than $\{X_n\}$ if and only if $\boldsymbol{\Sigma}_X(\mathbf{g}) \succeq \boldsymbol{\Sigma}_Y(\mathbf{g})$ for any vector-valued function $\mathbf{g}(\cdot) \in \mathbb{R}^d$. Consequently, in the UD-SGD framework, employing a more efficient sampling strategy $\{Y_n^i\}$ over the baseline $\{X_n^i\}$ by agent $i$ leads to $\boldsymbol{\Sigma}_{X^i}(\nabla F_i(\theta^*, \cdot)) \succeq \boldsymbol{\Sigma}_{Y^i}(\nabla F_i(\theta^*, \cdot))$, thus satisfying $\mathbf{U}_i^X \succeq \mathbf{U}_i^Y$. This finding, as per Corollary 3.4, implies an overall improvement in UD-SGD.

For illustration purposes, we list a few examples where two competing sampling strategies follow efficiency ordering: i) When an agent has complete access to the entire dataset (e.g., deep learning), shuffling techniques like single shuffling and random reshuffling are more efficient than *i.i.d.* sampling [38, 78]; ii) When an agent works with a graph-like data structure and employs a random walk, e.g., agent $i$ in Figure 1, using non-backtracking random walk (NBRW) is more efficient than simple random walk (SRW) [48]. iii) A recently proposed self-repellent random walk (SRRW) is shown to achieve *near-zero* sampling variance, indicating even higher sampling efficiency than NBRW and SRW [25].[3] This random-walk-based sampling finds a particular application in large-scale FL within D2D networks (e.g., mobile networks, wireless sensor networks), where each agent acts as an edge server or access point, gathering information from the local D2D network [37, 32]. Employing a random walk over local D2D network for each agent constitutes the sampling strategy.

Theorem 3.3 and Corollary 3.4 not only qualitatively compare these sampling strategies but also allow for a quantitative assessment of the overall system enhancement. Since every agent contributes equally to the limiting covariance matrix $\mathbf{V}$ of the distributed system as in Remark 3, a key application scenario is to encourage a subset of compliant agents to adopt highly efficient strategies like SRRW, potentially yielding better performance than universally upgrading to slightly improved strategies like NBRW. This approach, more feasible and impactful in large-scale machine learning scenarios where some agents cannot freely modify their sampling strategies, is a unique aspect of our framework not addressed in previous works focusing on the worst-performing agent [80, 42, 68, 72].

## 4 Experiments

In this section, we empirically evaluate the effect of agents' sampling strategies under various communication patterns in UD-SGD. We consider the $L_2$-regularized binary classification problem

$$\min_\theta f(\theta) \triangleq \frac{1}{N}\sum_{i=1}^N f_i(\theta), \text{ with } f_i(\theta) = \frac{1}{B}\sum_{j=1}^B \log\left(1+e^{\theta^T \mathbf{x}_{i,j}}\right) - y_{i,j}\left(\theta^T \mathbf{x}_{i,j}\right) + \frac{\kappa}{2}\|\theta\|^2, \quad (15)$$

where the feature vector $\mathbf{x}_{i,j}$ and its corresponding label $y_{i,j}$ are held by agent $i$, with a penalty parameter $\kappa$ set to 1. We use the *ijcnn1* dataset [14] with 22 features in each data point and 50k data points in total, which is evenly distributed to two groups with 50 agents each ($N = 100$ agents

---

[3]Note that SRRW is a *nonlinear* Markov chain that depends on the relative visit counts of each node in the graph. While its application in single-agent optimization has been studied in [39], expanding the theoretical examination of SRRW to multi-agent scenarios is beyond the scope of this paper. However, we can still numerically evaluate the performance of UD-SGD with multiple agents on general communication matrices using SRRW as a highly efficient sampling strategy in Section 4.

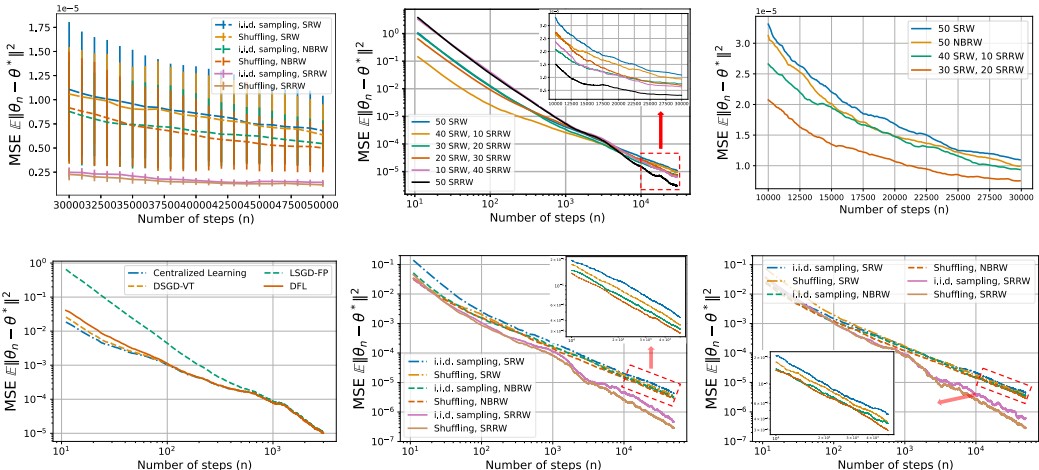

Figure 2: Binary classification problem. From left to right: (a) Impact of efficient sampling strategies on convergence. (b) Performance gains from partial adoption of efficient sampling. (c) Comparative advantage of SRRW over NBRW in a small subset of agents. (d) Asymptotic network independence of four algorithms under UD-SGD framework with fixed sampling strategy (shuffling, SRRW). (e) Different sampling strategies in the DSGD algorithm with time-varying topology (DSGD-VT). (f) Different sampling strategies in the DFL algorithm with increasing communication interval.

in total) and each agent holds $B = 500$ *distinct* data points. Each agent in the first group has full access to its entire dataset, and thus can employ *i.i.d.* sampling (baseline) or single shuffling. On the other hand, each agent in the other group has a graph-like structure and uses SRW (baseline), NBRW or SRRW with reweighting to sample its local dataset with uniform weight. In this simulation, we assume that agents can only communicate through a communication network using the DSGD algorithm. This scenario with heterogeneous agents, as depicted in Figure 1, is of great interest in large-scale machine learning [37, 32]. In addition, we employ a decreasing step size $\gamma_n = 1/n$ in our UD-SGD framework (1) because it is typically used for the strongly convex objective function and is tested to have the fastest convergence in this simulation setup. Due to space constraints, we defer detailed simulation setup, including the introduction of SRW, NBRW, and SRRW, to Appendix G.1.

The simulation results are obtained through 120 independent trials. In Figure 2(a), we assume that the first group of agents perform either *i.i.d.* sampling or shuffling method, while the other group of agents all change their sampling strategies from baseline SRW to NBRW and SRRW, as shown in the legend. This plot shows that improved sampling strategy leads to overall convergence speedup since NBRW and SRRW are more efficient than SRW [38, 25]. Furthermore, it illustrates that SRRW is significantly more efficient than NBRW in this simulation setup, i.e., **SRRW ≫ NBRW > SRW** in terms of sampling efficiency. While keeping the second group of agents unchanged, we can see that shuffling method outperforms *i.i.d.* sampling with smaller asymptotic MSE. However, shuffling method may not perform perfectly for small time $n$ due to slow mixing behavior in the initial period, which is also observed in the single-agent scenario in [65, 1, 38]. The error bar therein also indicates that the random-walk sampling strategy has a significant impact on the overall system performance and SRRW has smaller variance than NBRW and SRW.

In Figure 2(b), we let the first group of agents perform *i.i.d.* sampling while only changing a portion of agents in the second group to upgrade from SRW to SRRW, e.g., 30 SRW 20 SRRW in the legend means that there are 30 agents using SRW while the rest 20 agents in the second group upgrade to SRRW. We observe that more agents willing to upgrade from SRW to SRRW lead to smaller asymptotic MSE, as predicted by Theorem 3.3 and Remark 3. This improvement in MSE reduction doesn't scale linearly with more agents adopting SRRW because each agent holds its own dataset that are not necessarily identical, resulting in different individual limiting covariance matrices $\mathbf{V}_i \neq \mathbf{V}_j$.

While maintaining *i.i.d.* sampling for the first group of agents, we compare the performance when the second group of agents in Figure 2(c) employ NBRW or SRRW. Remarkably, the case with only 10 agents out of 50 agents in the second group adopting far more efficient sampling strategy (40 SRW, 10 SRRW) through incentives or compliance already produces a smaller MSE than all 50 agents using slightly better strategy (50 NBRW). The performance gap becomes even more pronounced

when 20 agents upgrade from SRW to SRRW (30 SRW, 20 SRRW). We show that the performance of a distributed system can be improved significantly when a small proportion of agents adopt highly efficient sampling strategies.

Figure 2(d) empirically illustrates the asymptotic network independence property via four algorithms under our UD-SGD framework: Centralized SGD (communication interval $K = 1$, communication matrix $\mathbf{W} = \mathbf{1}\mathbf{1}^T/N$); LSGD-FP (FL with full client participation, $K = 5$, $\mathbf{W} = \mathbf{1}\mathbf{1}^T/N$); DSGD-VT (DSGD with time-varying topologies, randomly chosen from 5 doubly stochastic matrices); DFL (decentralized FL with fixed MH-generated $\mathbf{W}$ and increasing communication interval $K_l = \max\{1, \log(l)\}$ after $l$-th aggregation). We fix the sampling strategy (shuffling, SRRW) throughout this plot. All four algorithms overlap around 1000 steps, implying that they have entered the asymptotic regime with similar performance where the CLT result dominates, implying the asymptotic network independence in the long run.

Figure 2(e) and 2(f) show the performance of different sampling strategies in DSGD-VT and DFL algorithms in terms of MSE. Both plots consistently demonstrate that improving agent's sampling strategies (e.g., shuffling > iid sampling, and SRRW > NBRW > SRW) leads to faster convergence with smaller MSE, supporting our theory.

Furthermore, in Appendix G.2, we simulate an image classification task with CIFAR-10 dataset [44] by training a 5-layer CNN and ResNet-18 model collaboratively through a 10-agent network. The result is illustrated in Figure 3, where SRRW outperforms NBRW and SRW as expected. In summary, we find that upgrading even a small portion of agents to efficient sampling strategies (e.g., shuffling method, NBRW, SRRW under different dataset structures) improves system performance in UD-SGD. These results are consistent in binary and image classification tasks, underscoring that *every agent matters* in distributed learning.

## 5 Conclusion

In this work, we develop an UD-SGD framework that establishes the CLT of various distributed algorithms with Markovian sampling. We overcome technical challenges such as quantifying consensus error under very general communication patterns and decomposing Markovian noise through the Poisson equation, which extends the analysis beyond the single-agent scenario. We demonstrate that even if only a few agents optimize their sampling strategies, the entire distributed system will benefit with a smaller limiting covariance in the CLT, suggesting a reduced MSE. This finding challenges the current established upper bounds where the worst-performing agent leads the pack. Future studies could pivot towards developing fine-grained finite-time bounds to individually characterize each agent's behavior, and theoretically analyze the effect of SRRW in UD-SGD.

## 6 Acknowledgments and Disclosure of Funding

We thank the anonymous reviewers for their constructive comments. This work was supported in part by National Science Foundation under Grant Nos. CNS-2007423, IIS-1910749, and IIS-2421484.

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

# A  Discussion of Assumption 2.3-ii)

## A.1  Suitable choices of $K_l$

When wet let $K_l \sim \log(l)$ (resp. $K_l \sim \log\log(l)$), as suggested by [51], it trivially satisfies $K_{\tau_n} = o(\gamma_n^{-1/2(L+1)}) = o(n^{1/2(L+1)})$ since by definition $K_{\tau_n} < K_n \sim \log(n)$ (resp. $\log\log(n)$), and $\log(n) = o(n^\epsilon)$ (resp. $\log\log(n) = o(n^\epsilon)$) for any $\epsilon > 0$. Besides, $\sum_n \eta_n^2 = \sum_n \gamma_n^2 K_{\tau_n}^{2(L+1)} \lesssim \sum_n n^{-2} n^{2(L+1)\epsilon} = \sum_n n^{2(L+1)\epsilon-2}$. To ensure $\sum_n \eta_n^2 < \infty$, it is sufficient to have $2(L+1)\epsilon - 2 < -1$, or equivalently, $\epsilon < 1/2(L+1)$. Since $\epsilon$ can be arbitrarily small to satisfy the condition, $\sum_n \eta_n^2 < \infty$ is satisfied. When $K_l \sim \log(l)$, we can rewrite the last condition as

$$
\frac{\eta_{n_l+1}}{\eta_{n_{l+1}+1}} = \frac{\gamma_{n_l+1}}{\gamma_{n_{l+1}+1}} \frac{K_l^{L+1}}{K_{l+1}^{L+1}} = \left( \frac{n_{l+1}+1}{n_l+1} \right) \left( \frac{\log(l+1)+1}{\log(l)+1} \right)^{L+1}
$$
$$
= \left( 1 + \frac{K_{l+1}}{n_l+1} \right) \left( \frac{\log(l+1)+1}{\log(l)+1} \right)^{L+1}, \tag{16}
$$

where we have $n_l \sim \log(l!)$ such that $K_{l+1}/n_l = \log(l+1)/\log(l!) \to 0$ and $\log(l+1)/\log(l) \to 1$ as $l \to \infty$, which leads to $\lim_{n\to\infty} \eta_{n_l+1}/\eta_{n_{l+1}+1} = 1$. Similarly, for $K_l \sim \log\log(l)$, we have $n_l \sim \log(\prod_{s=1}^l \log(s))$ such that $K_{l+1}/n_l \sim \log\log(l+1)/\log\log(\prod_{s=1}^l \log(s)) \to 0$ and $\log\log(l+1)/\log\log(l) \to 1$ as $l \to \infty$, which also leads to $\lim_{n\to\infty} \eta_{n_l+1}/\eta_{n_{l+1}+1} = 1$.

## A.2  Practical implications of Assumption 2.3-ii)

In this assumption, we allow the number of local iterations to go to infinity asymptotically. In distributed learning environments such as mobile, IoT, and wireless sensor networks, where nodes are often constrained by battery life, increasing communication interval in Assumption 2.3-ii) plays a crucial role in balancing energy costs with communication effectiveness. It allows agents to communicate more frequently early on, leading to a faster initial convergence to the neighborhood of $\theta^*$. Then, we slow down the communication frequency between agents to conserve energy, leveraging the diminishing returns on accuracy improvements from additional communications.

Consider the scenario where devices across multiple clusters collaborate on a distributed optimization task, utilizing local datasets. Devices within each cluster form a communication network that allows a virtual agent to perform a heterogeneous Markov chain trajectory via random walk, or an i.i.d. sequence in a complete graph with self-loops, depending on the application context. Each cluster features an edge server that supports the exchange of model estimates with neighboring clusters. By performing $K$ local updates before uploading these to the cluster's edge server, the model benefits from reduced communication overhead. As the frequency of updates between devices and edge servers decreases — optimized by gradually increasing $K$ — we effectively lower communication costs, particularly as the model estimation $\theta_n$ is close to $\theta^*$.

# B  Proof of Lemma 3.1

Let $\mathbf{J}_\perp \triangleq \mathbf{I}_N - \mathbf{J} \in \mathbb{R}^{N \times N}$ and $\mathcal{J}_\perp \triangleq \mathbf{J}_\perp \otimes \mathbf{I}_d \in \mathbb{R}^{Nd \times Nd}$, where $\otimes$ is the Kronecker product. Let $\Theta_n = [(\theta_n^1)^T, \cdots, (\theta_n^N)^T]^T \in \mathbb{R}^{Nd}$. Then, motivated by [58], we define a sequence $\phi_n \triangleq \eta_{n+1}^{-1} \mathcal{J}_\perp \Theta_n \in \mathbb{R}^{Nd}$ in the *increasing communication interval case* (resp. $\phi_n \triangleq \gamma_{n+1}^{-1} \mathcal{J}_\perp \Theta_n$ in the *bounded communication interval case*), where $\eta_{n+1}$ is defined in Assumption 2.3-ii). $\mathcal{J}_\perp \Theta_n = \Theta_n - \frac{1}{N}(\mathbf{11}^T \otimes \mathbf{I}_d)\Theta_n$ represents the consensus error of the model.

We first give the following lemma that shows the pathwise boundedness of $\phi_n$.

**Lemma B.1.** *Let Assumptions 2.1, 2.3, 2.4 and 2.5 hold. For any compact set $\Omega \subset \mathbb{R}^{Nd}$, the sequence $\phi_n$ satisfies $\sup_n \mathbb{E}[\|\phi_n\|^2 \mathbb{1}_{\cap_{j \leq n-1}\{\Theta_j \in \Omega\}}] < \infty$.*

Lemma B.1 and Assumption 2.3-ii) imply that for any $n \geq 0$, $\mathbb{E}[\|\mathcal{J}_\perp \Theta_n\|^2 \mathbb{1}_{\cap_{j \leq n-1}\{\Theta_j \in \Omega\}}] = \eta_{n+1}^2 \mathbb{E}[\|\phi_n\|^2 \mathbb{1}_{\cap_{j \leq n-1}\{\Theta_j \in \Omega\}}] \leq C\eta_{n+1}^2$ for some constant $C$ that depends on $C_1$ and $\Omega$. Along with Assumption 2.4 such that $\|\mathcal{J}_\perp \Theta_n\|$ is always bounded per each trajectory, it means

$$
\|\mathcal{J}_\perp \Theta_n\| \mathbb{1}_{\cap_{j \leq n-1}\{\Theta_j \in \Omega\}} = O(\eta_n) \quad a.s.
$$

Let $\{\Omega_m\}_{m\geq 0}$ be a sequence of increasing compact subset of $\mathbb{R}^{Nd}$ such that $\bigcup_m \Omega_m = \mathbb{R}^{Nd}$. Then, we know that for any $m \geq 0$,

$$\|\mathcal{J}_\perp \Theta_n\| \mathbb{1}_{\cap_{j\leq n-1}\{\Theta_j \in \Omega_m\}} = O(\eta_n) \quad a.s. \tag{17}$$

(17) indicates either one of the following two cases:

- there exists some trajectory-dependent index $m'$ such that each trajectory $\{\Theta_n\}_{n\geq 0}$ is always within the compact set $\Omega_{m'}$, i.e., $\mathbb{1}_{\cap_{j\leq n}\{\Theta_j \in \Omega_{m'}\}} = 1$ (satisfied by the construction of increasing compact sets $\{\Omega_m\}_{m\geq 0}$ and Assumption 2.4), and we have $\|\mathcal{J}_\perp \Theta_n\| = O(\eta_n)$ such that $\lim_{n\to\infty} \mathcal{J}_\perp \Theta_n = \mathbf{0}$;

- $\Theta_n$ will escape the compact set $\Omega_m$ eventually for any $m \geq 0$ in finite time such that $\mathbb{1}_{\cap_{j\leq n-1}\{\Theta_j \in \Omega_m\}} = 0$ when $n$ is large enough.

We can see the second case contradicts Assumption 2.4 because we assume every trajectory $\{\Theta_n\}_{n\geq 0}$ is within some compact set. Therefore, (17) for any $m \geq 0$ is equivalent to showing $\|\mathcal{J}_\perp \Theta_n\| = O(\eta_n)$ and $\lim_{n\to\infty} \mathcal{J}_\perp \Theta_n = \mathbf{0}$. Under Assumption 2.3-i) we can obtain similar result $\|\mathcal{J}_\perp \Theta_n\| = O(\gamma_n)$ by following the same steps as above, which completes the proof of Lemma 3.1.

*Proof of Lemma B.1.* We begin by rewriting (6) in the matrix form,

$$\Theta_{n+1} = \mathcal{W}_n \left( \Theta_n - \gamma_{n+1} \nabla \mathbf{F}(\Theta_n, \mathbf{X}_n) \right), \tag{18}$$

where $\mathbf{X}_n \triangleq (X_n^1, X_n^2, \cdots, X_n^N)$ and $\nabla \mathbf{F}(\Theta_n, \mathbf{X}_n) \triangleq [\nabla F_1(\theta_n^1, X_n^1)^T, \cdots, \nabla F_N(\theta_n^N, X_n^N)^T]^T \in \mathbb{R}^{Nd}$. Recall $\theta_n \triangleq \frac{1}{N}\sum_{i=1}^N \theta_n^i \in \mathbb{R}^d$ and we have $[\theta_n^T, \cdots, \theta_n^T]^T = \frac{1}{N}(\mathbf{1}\mathbf{1}^T \otimes \mathbf{I}_d)\Theta_n \in \mathbb{R}^{Nd}$.

**Case 1 (Increasing communication interval $K_{\tau_n}$):** By left multiplying (18) with $\frac{1}{N}(\mathbf{1}\mathbf{1}^T \otimes \mathbf{I}_d)$, along with $\gamma_{n+1} = \eta_{n+1}/K_{\tau_{n+1}}^{L+1}$ in Assumption 2.3-ii), we have the following iteration

$$\frac{1}{N}(\mathbf{1}\mathbf{1}^T \otimes \mathbf{I}_d)\Theta_{n+1} = \frac{1}{N}(\mathbf{1}\mathbf{1}^T \otimes \mathbf{I}_d)\Theta_n - \eta_{n+1}\frac{1}{N}(\mathbf{1}\mathbf{1}^T \otimes \mathbf{I}_d)\frac{\nabla \mathbf{F}(\Theta_n, \mathbf{X}_n)}{K_{\tau_{n+1}}^{L+1}}, \tag{19}$$

where the equality comes from $\frac{1}{N}(\mathbf{1}\mathbf{1}^T \otimes \mathbf{I}_d)\mathcal{W}_n = \frac{1}{N}(\mathbf{1}\mathbf{1}^T \mathbf{W}_n \otimes \mathbf{I}_d) = \frac{1}{N}(\mathbf{1}\mathbf{1}^T \otimes \mathbf{I}_d)$. With (18) and (19), we have

$$\begin{aligned}
&\Theta_{n+1} - \frac{1}{N}(\mathbf{1}\mathbf{1}^T \otimes \mathbf{I}_d)\Theta_{n+1} \\
&= \left(\mathcal{W}_n - \frac{1}{N}(\mathbf{1}\mathbf{1}^T \otimes \mathbf{I}_d)\right)\Theta_n - \eta_{n+1}\left(\mathcal{W}_n - \frac{1}{N}(\mathbf{1}\mathbf{1}^T \otimes \mathbf{I}_d)\right)\frac{\nabla \mathbf{F}(\Theta_n, \mathbf{X}_n)}{K_{\tau_{n+1}}^{L+1}} \\
&= (\mathbf{J}_\perp \mathbf{W}_n \otimes \mathbf{I}_d)\mathcal{J}_\perp \Theta_n - \eta_{n+1}(\mathbf{J}_\perp \mathbf{W}_n \otimes \mathbf{I}_d)\frac{\nabla \mathbf{F}(\Theta_n, X_n)}{K_{\tau_{n+1}}^{L+1}} \\
&= \eta_{n+1}(\mathbf{J}_\perp \mathbf{W}_n \otimes \mathbf{I}_d)\left(\eta_{n+1}^{-1}\mathcal{J}_\perp \Theta_n - \frac{\nabla \mathbf{F}(\Theta_n, \mathbf{X}_n)}{K_{\tau_{n+1}}^{L+1}}\right),
\end{aligned} \tag{20}$$

where the second equality comes from $\mathcal{W}_n - \frac{1}{N}(\mathbf{1}\mathbf{1}^T \otimes \mathbf{I}_d) = (\mathbf{W}_n - \frac{1}{N}\mathbf{1}\mathbf{1}^T) \otimes \mathbf{I}_d = \mathbf{J}_\perp \mathbf{W}_n \otimes \mathbf{I}_d$ and $(\mathbf{J}_\perp \mathbf{W}_n \otimes \mathbf{I}_d)\mathcal{J}_\perp = \mathbf{J}_\perp \mathbf{W}_n \mathbf{J}_\perp \otimes \mathbf{I}_d = \mathbf{J}_\perp \mathbf{W}_n \otimes \mathbf{I}_d$. Let $a_n \triangleq \eta_n/\eta_{n+1}$, dividing both sides of (20) by $\eta_{n+2}$ gives

$$\phi_{n+1} = a_{n+1}(\mathbf{J}_\perp \mathbf{W}_n \otimes \mathbf{I}_d)\left(\phi_n - \frac{\nabla \mathbf{F}(\Theta_n, \mathbf{X}_n)}{K_{\tau_{n+1}}^{L+1}}\right). \tag{21}$$

Define the filtration $\{\mathcal{F}_n\}_{n\geq 0}$ as $\mathcal{F}_n \triangleq \sigma\{\Theta_0, \mathbf{X}_0, \mathbf{W}_0, \Theta_1, \mathbf{X}_1, \mathbf{W}_1, \cdots, \mathbf{X}_{n-1}, \mathbf{W}_{n-1}, \Theta_n, \mathbf{X}_n\}$. Recursively computing (21) w.r.t the time interval $[n_l, n_{l+1}]$ gives

$$
\begin{aligned}
\phi_{n_{l+1}} &= \left[\prod_{k=n_l+1}^{n_{l+1}} a_k\right] \left(\left[\mathbf{J}_\perp \prod_{k=n_l}^{n_{l+1}-1} \mathbf{W}_k\right] \otimes \mathbf{I}_d\right) \phi_{n_l} \\
&\quad - \sum_{k=n_l}^{n_{l+1}-1} \left[\prod_{i=k+1}^{n_{l+1}} a_i\right] \left(\left[\mathbf{J}_\perp \prod_{i=k}^{n_{l+1}-1} \mathbf{W}_i\right] \otimes \mathbf{I}_d\right) \frac{\nabla\mathbf{F}(\Theta_k, \mathbf{X}_k)}{K_{l+1}^{L+1}} \\
&= \frac{\eta_{n_l+1}}{\eta_{n_{l+1}+1}} \left(\mathbf{J}_\perp \mathbf{W}_{n_l} \otimes \mathbf{I}_d\right) \phi_{n_l} - \sum_{k=n_l}^{n_{l+1}-1} \frac{\eta_{n_l+1}}{\eta_{k+2}} \left(\mathbf{J}_\perp \mathbf{W}_{n_l} \otimes \mathbf{I}_d\right) \frac{\nabla\mathbf{F}(\Theta_k, \mathbf{X}_k)}{K_{l+1}^{L+1}},
\end{aligned}
\tag{22}
$$

where $\prod$ is the backward multiplier, the second equality comes from $\mathbf{J}_\perp \mathbf{W}_n \mathbf{J}_\perp = \mathbf{J}_\perp \mathbf{W}_n$ and $\mathbf{W}_k = \mathbf{I}_N$ for $k \notin \{n_l\}$. In Assumption 2.5, we have $\|\mathbb{E}_{\mathbf{W}\sim\mathcal{P}_{n_l}}[\mathbf{W}^T\mathbf{J}_\perp\mathbf{W}]\| = \|\mathbb{E}_{\mathbf{W}\sim\mathcal{P}_{n_l}}[\mathbf{W}^T\mathbf{W} - \mathbf{J}]\| \leq C_1 < 1$. Then,

$$
\begin{aligned}
&\mathbb{E}[\|\phi_{n_{l+1}}\|^2 | \mathcal{F}_{n_l}] \\
&= \left(\frac{\eta_{n_l+1}}{\eta_{n_{l+1}+1}}\right)^2 \phi_{n_l}^T \mathbb{E}_{\mathbf{W}_{n_l}\sim\mathcal{P}_{n_l}} \left[\left(\mathbf{J}_\perp \mathbf{W}_{n_l} \otimes \mathbf{I}_d\right)^T \left(\mathbf{J}_\perp \mathbf{W}_{n_l} \otimes \mathbf{I}_d\right)\right] \phi_{n_l} \\
&\quad - 2\mathbb{E}\left[\sum_{k=n_l}^{n_{l+1}-1} \frac{\eta_{n_l+1}^2}{\eta_{n_{l+1}+1}\eta_{k+2}} \phi_{n_l}^T \left(\mathbf{J}_\perp \mathbf{W}_{n_l} \otimes \mathbf{I}_d\right)^T \left(\mathbf{J}_\perp \mathbf{W}_{n_l} \otimes \mathbf{I}_d\right) \frac{\nabla\mathbf{F}(\Theta_k, \mathbf{X}_k)}{K_{l+1}^{L+1}} \middle| \mathcal{F}_{n_l}\right] \\
&\quad + \mathbb{E}\left[\left\|\sum_{k=n_l}^{n_{l+1}-1} \frac{\eta_{n_l+1}}{\eta_{k+2}} \left(\mathbf{J}_\perp \mathbf{W}_{n_l} \otimes \mathbf{I}_d\right) \frac{\nabla\mathbf{F}(\Theta_k, \mathbf{X}_k)}{K_{l+1}^{L+1}}\right\|^2 \middle| \mathcal{F}_{n_l}\right] \\
&\leq \left(\frac{\eta_{n_l+1}}{\eta_{n_{l+1}+1}}\right)^2 \phi_{n_l}^T \mathbb{E}_{\mathbf{W}_{n_l}\sim\mathcal{P}_{n_l}} \left[\left(\mathbf{W}_{n_l}^T \mathbf{J}_\perp \mathbf{W}_{n_l} \otimes \mathbf{I}_d\right)\right] \phi_{n_l} \\
&\quad - 2\left(\frac{\eta_{n_l+1}}{\eta_{n_{l+1}+1}}\right)^2 \mathbb{E}\left[\sum_{k=n_l}^{n_{l+1}-1} \phi_{n_l}^T \left(\mathbf{W}_{n_l}^T \mathbf{J}_\perp \mathbf{W}_{n_l} \otimes \mathbf{I}_d\right) \frac{\nabla\mathbf{F}(\Theta_k, \mathbf{X}_k)}{K_{l+1}^{L+1}} \middle| \mathcal{F}_{n_l}\right] \\
&\quad + \left(\frac{\eta_{n_l+1}}{\eta_{n_{l+1}+1}}\right)^2 \mathbb{E}\left[\left\|\left(\mathbf{J}_\perp \mathbf{W}_{n_l} \otimes \mathbf{I}_d\right) \sum_{k=n_l}^{n_{l+1}-1} \frac{\nabla\mathbf{F}(\Theta_k, \mathbf{X}_k)}{K_{l+1}^{L+1}}\right\|^2 \middle| \mathcal{F}_{n_l}\right] \\
&\leq \left(\frac{\eta_{n_l+1}}{\eta_{n_{l+1}+1}}\right)^2 C_1 \|\phi_{n_l}\|^2 + 2\left(\frac{\eta_{n_l+1}}{\eta_{n_{l+1}+1}}\right)^2 C_1 \|\phi_{n_l}\| \mathbb{E}\left[\left\|\sum_{k=n_l}^{n_{l+1}-1} \frac{\nabla\mathbf{F}(\Theta_k, \mathbf{X}_k)}{K_{l+1}^{L+1}}\right\| \middle| \mathcal{F}_{n_l}\right] \\
&\quad + \left(\frac{\eta_{n_l+1}}{\eta_{n_{l+1}+1}}\right)^2 C_1 \mathbb{E}\left[\left\|\sum_{k=n_l}^{n_{l+1}-1} \frac{\nabla\mathbf{F}(\Theta_k, \mathbf{X}_k)}{K_{l+1}^{L+1}}\right\|^2 \middle| \mathcal{F}_{n_l}\right],
\end{aligned}
\tag{23}
$$

where the first inequality comes from $\mathbf{J}_\perp^T \mathbf{J}_\perp = \mathbf{J}_\perp$ and $\eta_{k+2} \geq \eta_{n_{l+1}+1}$ for $k \in [n_l, n_{l+1} - 1]$. Then, we analyze the norm of the gradient $\|\nabla\mathbf{F}(\Theta_k, \mathbf{X}_k)\|$ in the second term on the RHS of (23) conditioned on $\mathcal{F}_{n_l}$. By Assumption 2.4, we assume $\Theta_{n_l}$ is within some compact set $\Omega$ at time $n_l$ such that $\sup_{i\in[N], X^i\in\mathcal{X}_i} \nabla F_i(\theta_{n_l}^i, X^i) \leq C_\Omega$ for some constant $C_\Omega$. For $n = n_l + 1$ and any $\mathbf{X} \in \mathcal{X}_1 \times \mathcal{X}_2 \times \cdots \times \mathcal{X}_N$, we have

$$
\|\nabla\mathbf{F}(\Theta_{n_l+1}, \mathbf{X})\| \leq \|\nabla\mathbf{F}(\Theta_{n_l+1}, \mathbf{X}) - \nabla\mathbf{F}(\Theta_{n_l}, \mathbf{X})\| + \|\nabla\mathbf{F}(\Theta_{n_l}, \mathbf{X})\|.
$$

Considering $\|\nabla\mathbf{F}(\Theta_{n_l}, \mathbf{X})\|$, we have $\sup_{\mathbf{X}} \|\nabla\mathbf{F}(\Theta_{n_l}, \mathbf{X})\|^2 \leq \sum_{i=1}^{N} \sup_{X^i\in\mathcal{X}_i} \|\nabla F_i(\theta_{n_l}^i, X^i)\|^2 \leq NC_\Omega^2$ such that $\|\nabla\mathbf{F}(\Theta_{n_l}, \mathbf{X})\| \leq \sqrt{N}C_\Omega$. In addition, we

have

$$\|\nabla\mathbf{F}(\Theta_{n_l+1},\mathbf{X}) - \nabla\mathbf{F}(\Theta_{n_l},\mathbf{X})\|^2 = \sum_{i=1}^{N}\|\nabla F_i(\theta_{n_l+1}^i, X^i) - \nabla F_i(\theta_{n_l}^i, X^i)\|^2$$

$$\leq \sum_{i=1}^{N} L^2\|\theta_{n_l+1}^i - \theta_{n_l}^i\|^2 \qquad (24)$$

$$\leq \sum_{i=1}^{N}\gamma_{n_l+1}^2 L^2\|\nabla F_i(\theta_{n_l}^i, X_{n_l}^i)\|^2$$

$$\leq \gamma_{n_l+1}^2 C_\Omega^2 N L^2$$

such that $\|\nabla\mathbf{F}(\Theta_{n_l+1},\mathbf{X}) - \nabla\mathbf{F}(\Theta_{n_l},\mathbf{X})\| \leq \gamma_{n_l+1}C_\Omega\sqrt{N}L$. Thus, for any $\mathbf{X}$,

$$\|\nabla\mathbf{F}(\Theta_{n_l+1},\mathbf{X})\| \leq (1 + \gamma_{n_l+1}L)\sqrt{N}C_\Omega. \qquad (25)$$

For $n = n_l + 2$ and any $\mathbf{X}$, we have

$$\|\nabla\mathbf{F}(\Theta_{n_l+2},\mathbf{X})\| \leq \|\nabla\mathbf{F}(\Theta_{n_l+2},\mathbf{X}) - \nabla\mathbf{F}(\Theta_{n_l+1},\mathbf{X})\| + \|\nabla\mathbf{F}(\Theta_{n_l+1},\mathbf{X})\|.$$

Similar to the steps in (24), we have

$$\|\nabla\mathbf{F}(\Theta_{n_l+2},\mathbf{X}) - \nabla\mathbf{F}(\Theta_{n_l+1},\mathbf{X})\|^2 \leq \sum_{i=1}^{N}\gamma_{n_l+2}^2 L^2\|\nabla F_i(\theta_{n_l+1}^i, X_{n_l+1}^i)\|^2 \qquad (26)$$

$$= \gamma_{n_l+2}^2 L^2\|\nabla\mathbf{F}(\Theta_{n_l+1},\mathbf{X}_{n_l+1})\|^2.$$

Then, $\|\nabla\mathbf{F}(\Theta_{n_l+2},\mathbf{X})\| \leq (1 + \gamma_{n_l+2}L)\sup_\mathbf{X}\|\nabla\mathbf{F}(\Theta_{n_l+1},\mathbf{X})\|$ and, together with (25), we have

$$\|\nabla\mathbf{F}(\Theta_{n_l+2},\mathbf{X})\| \leq (1 + \gamma_{n_l+2}L)(1 + \gamma_{n_l+1}L)\sqrt{N}C_\Omega. \qquad (27)$$

By induction, $\|\nabla\mathbf{F}(\Theta_{n_l+m},\mathbf{X})\| \leq \prod_{s=1}^{m}(1 + \gamma_{n_l+s}L)\sqrt{N}C_\Omega$ for $m \in [1, K_{l+1} - 1]$.

The next step is to analyze the growth rate of $\prod_{s=1}^{m}(1 + \gamma_{n_l+s}L)$. By $1 + x \leq e^x$ for $x \geq 0$, we have

$$\prod_{s=1}^{m}(1 + \gamma_{n_l+s}L) \leq e^{L\sum_{s=1}^{m}\gamma_{n_l+s}}.$$

For step size $\gamma_n = 1/n$, we have $L\sum_{s=1}^{m}\gamma_{n_l+s} = L\sum_{s=1}^{m}1/(n_l+s) < L\sum_{s=1}^{m}1/s < L(\log(m)+1)$ such that $\prod_{s=1}^{m}(1 + \gamma_{n_l+s}L) < (em)^L$. Then,

$$\left\|\sum_{k=n_l}^{n_{l+1}-1}\frac{\nabla\mathbf{F}(\Theta_k,\mathbf{X}_k)}{K_{l+1}^{L+1}}\right\| \leq \frac{1}{K_{l+1}^{L+1}}\sum_{k=n_l}^{n_{l+1}-1}\|\nabla\mathbf{F}(\Theta_k,\mathbf{X}_k)\| \leq \frac{1}{K_{l+1}^{L+1}}\sqrt{N}e^L C_\Omega\sum_{m=0}^{K_{l+1}-1}m^L \qquad (28)$$

$$\leq \sqrt{N}e^L C_\Omega,$$

where the last inequality comes from $\sum_{m=0}^{K_{l+1}-1}m^L < K_{l+1}(K_{l+1}-1)^L < K_{l+1}^{L+1}$. We can see the sum of the norm of the gradients are bounded by $\sqrt{N}e^L C_\Omega$, which only depends on the compact set $\Omega$ at time $n = n_l$.

Let $\delta_1 \in (C_1, 1)$. Since from Assumption 2.3-ii), $\lim_{l\to\infty}\eta_{n_l+1}/\eta_{n_{l+1}+1} = 1$, there exists some large enough $l_0$ such that $(\frac{\eta_{n_l+1}}{\eta_{n_{l+1}+1}})^2 C_1 < \delta_1 < \delta_2 := (\delta_1 + 1)/2 < 1$ for any $l > l_0$. Note that $\delta_1$ depends only on $C_1$ and is independent of $\mathcal{F}_n$. Then, let $\tilde{C}_\Omega := \sqrt{N}e^L C_\Omega$, we can rewrite (23) as

$$\mathbb{E}[\|\phi_{n_l+1}\|^2|\mathcal{F}_{n_l}] \leq \delta_1\|\phi_{n_l}\|^2 + 2\delta_1\tilde{C}_\Omega\|\phi_{n_l}\| + \delta_1\tilde{C}_\Omega^2 \qquad (29)$$

$$\leq \delta_2\|\phi_{n_l}\|^2 + M_\Omega,$$

where $M_\Omega$ satisfies $M_\Omega > 8\tilde{C}_\Omega^2/(1 - \delta_1) + \delta_1\tilde{C}_\Omega^2$, which is derived from rearranging (29) as $M_\Omega \geq (\delta_1 - \delta_2)\|\phi_{n_l}\|^2 + 2\delta_1\tilde{C}_\Omega\|\phi_{n_l}\| + \delta_1\tilde{C}_\Omega^2$ and upper bounding the RHS. Upon noting that $\mathbb{1}_{\cap_{j\leq n_l}\{\Theta_j\in\Omega\}} \leq \mathbb{1}_{\cap_{j\leq n_{l-1}}\{\Theta_j\in\Omega\}}$, we obtain

$$\mathbb{E}\left[\|\phi_{n_l+1}\|^2\mathbb{1}_{\cap_{j\leq n_l}\{\Theta_j\in\Omega\}}\right] \leq \delta_2\mathbb{E}\left[\|\phi_{n_l}\|^2\mathbb{1}_{\cap_{j\leq n_{l-1}}\{\Theta_j\in\Omega\}}\right] + M_\Omega. \qquad (30)$$

The induction leads to $\mathbb{E}[\|\phi_{n_{l+1}}\|^2 \mathbb{1}_{\cap_{j \leq n_l}\{\Theta_j \in \Omega\}}] \leq \delta_2^{n_{l+1}-n_{l_0}} \mathbb{E}[\|\phi_{n_{l_0}}\|^2 \mathbb{1}_{\cap_{j \leq n_{l_0}-1}\{\Theta_j \in \Omega\}}] +$ $M/(1-\delta_2) < \infty$ for any $l \geq l_0$. Besides, for $m \in (n_l, n_{l+1})$, by following the above steps (23) applied to (21), we have

$$\mathbb{E}[\|\phi_m\|^2 | \mathcal{F}_{n_l}] \leq \left(\frac{\eta_{n_l+1}}{\eta_{m+1}}\right)^2 \|\phi_{n_l}\|^2 + 2\left(\frac{\eta_{n_l+1}}{\eta_{m+1}}\right)^2 \|\phi_{n_l}\| \mathbb{E}\left[\left\|\sum_{k=n_l}^{m-1} \frac{\nabla \mathbf{F}(\Theta_k, \mathbf{X}_k)}{K_{l+1}^{L+1}}\right\| \Bigg| \mathcal{F}_{n_l}\right]$$
$$+ \left(\frac{\eta_{n_l+1}}{\eta_{m+1}}\right)^2 \mathbb{E}\left[\left\|\sum_{k=n_l}^{m-1} \frac{\nabla \mathbf{F}(\Theta_k, \mathbf{X}_k)}{K_{l+1}^{L+1}}\right\|^2 \Bigg| \mathcal{F}_{n_l}\right]. \tag{31}$$

By (28) we already show that $\|\sum_{k=n_l}^{n_{l+1}-1} \frac{\nabla \mathbf{F}(\Theta_k, \mathbf{X}_k)}{K_{l+1}^{L+1}}\| < \infty$ conditioned on $\mathcal{F}_{n_l}$. Therefore, $\mathbb{E}[\|\phi_m\|^2 \mathbb{1}_{\cap_{j \leq n_l}\{\Theta_j \in \Omega\}}] < \infty$ for $m \in (n_l, n_{l+1})$. This completes the boundedness analysis of $\mathbb{E}[\|\phi_n\|^2 \mathbb{1}_{\cap_{j \leq n-1}\{\Theta_j \in \Omega\}}]$.

**Case 2 (Bounded communication interval $K_{\tau_n} \leq K$):** In this case, we do not need the auxiliary step size $\eta_n$ and can directly work on $\gamma_n = 1/n^a$ for $a \in (0.5, 1]$. Similar to (20), we have

$$\Theta_{n+1} - \frac{1}{N}(\mathbf{1}\mathbf{1}^T \otimes \mathbf{I}_d)\Theta_{n+1} = \gamma_{n+1}(\mathbf{J}_\perp \mathbf{W}_n \otimes \mathbf{I}_d)\left(\gamma_{n+1}^{-1} \mathcal{J}_\perp \Theta_n - \nabla \mathbf{F}(\Theta_n, \mathbf{X}_n)\right), \tag{32}$$

and let $b_n \triangleq \gamma_n/\gamma_{n+1}$, dividing both sides of above equation by $\gamma_{n+2}$ gives

$$\phi_{n+1} = b_{n+1}(\mathbf{J}_\perp \mathbf{W}_n \otimes \mathbf{I}_d)(\phi_n - \nabla \mathbf{F}(\Theta_n, \mathbf{X}_n)). \tag{33}$$

Then, by following the similar steps in (22) and (23), we obtain

$$\mathbb{E}[\|\phi_{n_{l+1}}\|^2 | \mathcal{F}_{n_l}] \leq \left(\frac{\gamma_{n_l+1}}{\gamma_{n_{l+1}+1}}\right)^2 C_1\left(\|\phi_{n_l}\|^2 + 2\|\phi_{n_l}\| \mathbb{E}\left[\left\|\sum_{k=n_l}^{n_{l+1}-1} \nabla \mathbf{F}(\Theta_k, \mathbf{X}_k)\right\| \Bigg| \mathcal{F}_{n_l}\right]\right.$$
$$\left. + \mathbb{E}\left[\left\|\sum_{k=n_l}^{n_{l+1}-1} \nabla \mathbf{F}(\Theta_k, \mathbf{X}_k)\right\|^2 \Bigg| \mathcal{F}_{n_l}\right]\right). \tag{34}$$

Also similar to (25) - (28), we can bound the sum of the norm of the gradients as

$$\left\|\sum_{k=n_l}^{n_{l+1}-1} \nabla \mathbf{F}(\Theta_k, \mathbf{X}_k)\right\| \leq \sum_{k=n_l}^{n_{l+1}-1} \left[\prod_{s=n_l}^{k} (1 + \gamma_{s+1}L)\right] \sqrt{N} C_\Omega. \tag{35}$$

Now that $K_l$ is bounded above by $K$, $\prod_{s=n_l}^{k}(1 + \gamma_{s+1}L) \leq e^{L \sum_{s=n_l}^{k} \gamma_{s+1}} < e^{L \sum_{s=0}^{K-1} \gamma_{s+1}} := C_K$. Then, we further bound (35) as

$$\left\|\sum_{k=n_l}^{n_{l+1}-1} \nabla \mathbf{F}(\Theta_k, \mathbf{X}_k)\right\| \leq \sqrt{N} K C_K C_\Omega. \tag{36}$$

The subsequent proof is basically a replication of (29) - (31) and is therefore omitted. $\qquad\square$

## C  Proof of Theorem 3.2

We focus on analyzing the convergence property of $\theta$, which is obtained by left multiplying (18) with $\frac{1}{N}(\mathbf{1}^T \otimes \mathbf{I}_d)$, i.e.,

$$\theta_{n+1} = \frac{1}{N}(\mathbf{1}^T \otimes \mathbf{I}_d)\theta_{n+1}$$
$$= \theta_n - \gamma_{n+1}\frac{1}{N}(\mathbf{1}^T \otimes \mathbf{I}_d)\nabla \mathbf{F}(\Theta_n, \mathbf{X}_n). \tag{37}$$

where the second equality comes from $\mathbf{W}_n$ being doubly stochastic and $\frac{1}{N}(\mathbf{1}^T \otimes \mathbf{I}_d)\mathcal{W}_n = \frac{1}{N}(\mathbf{1}^T \mathbf{W}_n \otimes \mathbf{I}_d) = \frac{1}{N}(\mathbf{1}^T \otimes \mathbf{I}_d)$.

For self-contained purpose, we first give the almost sure convergence result for the stochastic approximation that will be used in our proof.

**Theorem C.1** (Theorem 2 [23]). *Consider the stochastic approximation in the form of*

$$\theta_{n+1} = \theta_n + \gamma_{n+1} h(\theta_n) + \gamma_{n+1} e_{n+1} + \gamma_{n+1} r_{n+1}. \tag{38}$$

*Assume that*

C1. *w.p.1, the closure of $\{\theta_n\}_{n \geq 0}$ is a compact subset of $\mathbb{R}^d$;*

C2. *$\{\gamma_n\}$ is a decreasing sequence of positive number such that $\sum_n \gamma_n = \infty$;*

C3. *w.p.1, $\lim_{p \to \infty} \sum_{n=1}^{p} \gamma_n e_n$ exists and is finite. Moreover, $\lim_{n \to \infty} r_n = 0$.*

C4. *vector-valued function $h$ is continuous on $R^d$ and there exists a continuously differentiable function $V : \mathbb{R}^d \to \mathbb{R}$ such that $\langle \nabla V(\theta), h(\theta) \rangle \leq 0$ for all $\theta \in \mathbb{R}^d$. Besides, the interior of $V(\mathcal{L})$ is empty where $\mathcal{L} \triangleq \{\theta \in \mathbb{R}^d : \langle \nabla V(\theta), h(\theta) \rangle = 0\}$.*

*Then, w.p.1, $\limsup_n d(\theta_n, \mathcal{L}) = 0$.* $\qquad\qquad\square$

We can rewrite (37) as

$$\begin{aligned}
\theta_{n+1} =& \theta_n - \gamma_{n+1} \frac{1}{N} (\mathbf{1}^T \otimes \mathbf{I}_d) \nabla \mathbf{F}(\Theta_n, \mathbf{X}_n) \\
=& \theta_n - \gamma_{n+1} \nabla f(\theta_n) - \gamma_{n+1} \left( \frac{1}{N} \sum_{i=1}^{N} \nabla f_i(\theta_n^i) - \nabla f(\theta_n) \right) \\
& - \gamma_{n+1} \left( \frac{1}{N} \sum_{i=1}^{N} \nabla F_i(\theta_n^i, X_n^i) - \frac{1}{N} \sum_{i=1}^{N} \nabla f_i(\theta_n^i) \right),
\end{aligned} \tag{39}$$

and work on the converging behavior of the third and fourth term. By definition of function $\nabla f(\cdot)$, we have

$$r_n \triangleq \frac{1}{N} \sum_{i=1}^{N} \nabla f_i(\theta_n^i) - \nabla f(\theta_n) = \frac{1}{N} \sum_{i=1}^{N} \left[ \nabla f_i(\theta_n^i) - \nabla f_i(\theta_n) \right]. \tag{40}$$

By the Lipschitz continuity of function $\nabla F_i(\cdot, X)$ in (7), we have

$$\|r_n\| \leq \frac{1}{N} \sum_{i=1}^{N} L \|\theta_n^i - \theta_n\| \leq \frac{L}{\sqrt{N}} \left\| \Theta_n - \frac{1}{N} (\mathbf{1}\mathbf{1}^T \otimes \mathbf{I}_d) \Theta_n \right\| = \frac{L}{\sqrt{N}} \|\mathcal{J}_\perp \Theta_n\|, \tag{41}$$

where the second inequality comes from the Cauchy-Schwartz inequality. In Appendix B, we have shown $\lim_n \mathcal{J}_\perp \Theta_n = \mathbf{0}$ almost surely such that $\lim_{n \to \infty} r_n = 0$ almost surely.

Next, we further decompose the fourth term in (39). For an ergodic transition matrix $\mathbf{P}$ and a function $v$ associated with the same state space $\mathcal{X}$, define the operator $\mathbf{P}^k v(x) \triangleq \sum_{y \in \mathcal{X}} \mathbf{P}^k(x, y) v(y)$ for the $k$-step transition probability $\mathbf{P}^k(x, y)$. Denote by $\mathbf{P}_1, \cdots, \mathbf{P}_N$ the underlying transition matrices of all $N$ agents with corresponding stationary distribution $\boldsymbol{\pi}_1, \cdots, \boldsymbol{\pi}_N$. Then, for every function $\nabla F_i(\theta^i, \cdot) : \mathcal{X}_i \to \mathbb{R}^d$, there exists a corresponding function $m_{\theta^i}(\cdot) : \mathcal{X}_i \to \mathbb{R}^d$ such that

$$m_{\theta^i}(x) - \mathbf{P}_i m_{\theta^i}(x) = \nabla F_i(\theta^i, x) - \nabla f_i(\theta^i). \tag{42}$$

The solution of the Poisson equation (42) has been studied in the literature, e.g., [17, 38]. For self-contained purpose, we derive the closed-form $m_{\theta^i}(x)$ from scratch. First of all, we can obtain function $m_{\theta^i}(x)$ in the recursive form as follows,

$$m_{\theta^i}(x) = \nabla F_i(\theta^i, x) - \nabla f_i(\theta^i) + \mathbf{P}_i [\nabla F_i(\theta^i, \cdot) - \nabla f_i(\theta^i)](x) + \mathbf{P}_i^2 [\nabla F_i(\theta^i, \cdot) - \nabla f_i(\theta^i)](x) + \cdots. \tag{43}$$

It is not hard to check that (43) satisfies (42). Note that by induction we get

$$\mathbf{P}_i^k - \mathbf{1}(\boldsymbol{\pi}_i)^T = \left( \mathbf{P}_i - \mathbf{1}(\boldsymbol{\pi}_i)^T \right)^k, \forall k \in \mathbb{N}, k \geq 1. \tag{44}$$

Then, we can further simplify (43), and the closed-form expression of $m_{\theta^i}(x)$ is given as

$$
\begin{aligned}
m_{\theta^i}(x) &= \sum_{y \in \mathcal{X}_i} \left[ \mathbf{P}_i - \mathbf{1}(\boldsymbol{\pi}_i)^T \right]^0 (x,y)(\nabla F_i(\theta^i, y) - \nabla f_i(\theta^i)) \\
&\quad + \sum_{y \in \mathcal{X}^i} \left[ \mathbf{P}_i^1 - \mathbf{1}(\boldsymbol{\pi}_i)^T \right] (x,y)(\nabla F_i(\theta^i, y) - \nabla f_i(\theta^i)) + \cdots \\
&= \sum_{y \in \mathcal{X}_i} \left[ \sum_{k=0}^{\infty} \left[ \mathbf{P}_i - \mathbf{1}(\boldsymbol{\pi}_i)^T \right]^k \right] (x,y)(\nabla F_i(\theta^i, y) - \nabla f_i(\theta^i)) \\
&= \sum_{y \in \mathcal{X}_i} \left( \mathbf{I} - \mathbf{P}_i + \mathbf{1}(\boldsymbol{\pi}_i)^T \right)^{-1} (x,y)(\nabla F_i(\theta^i, y) - \nabla f_i(\theta^i)),
\end{aligned}
\tag{45}
$$

where the fourth equality comes from (44). Note that the so-called 'fundamental matrix' $(\mathbf{I} - \mathbf{P}_i + \mathbf{1}(\boldsymbol{\pi}_i)^T)^{-1}$ exists for every ergodic Markov chain $X^i$ from Assumption 2.2. Since function $\nabla F_i$ is Lipschitz continuous, we have the following lemma.

**Lemma C.2.** *Under assumption (A1), functions $m_{\theta^i}(x)$ and $\mathbf{P}_i m_{\theta^i}(x)$ are both Lipschitz continuous in $\theta^i$ for any $x \in \mathcal{X}_i$.*

*Proof.* By (45), for any $\theta_1^i, \theta_2^i \in \mathbb{R}^d$ and $x \in \mathcal{X}_i$, we have

$$
\begin{aligned}
\left\| m_{\theta_1^i}(x) - m_{\theta_2^i}(x) \right\| &\leq \left\| \sum_{y \in \mathcal{X}_i} \left( \mathbf{I} - \mathbf{P}_i + \mathbf{1}(\boldsymbol{\pi}_i)^T \right)^{-1} (x,y) \left[ \nabla F_i(\theta_1^i, y) - \nabla F_i(\theta_2^i, y) \right] \right\| \\
&\quad + \left\| \nabla f_i(\theta_1^i) - \nabla f_i(\theta_2^i) \right\| \\
&\leq C_i \max_{y \in \mathcal{X}_i} \left\| \nabla F_i(\theta_1^i, y) - \nabla F_i(\theta_2^i, y) \right\| + \left\| \nabla f_i(\theta_1^i) - \nabla f_i(\theta_2^i) \right\| \\
&\leq (C_i L + 1) \| \theta_1^i - \theta_2^i \|,
\end{aligned}
\tag{46}
$$

where the second inequality holds for a constant $C_i$ that is the largest absolute value of the entry in the matrix $(\mathbf{I} - \mathbf{P}_i + \mathbf{1}(\boldsymbol{\pi}_i)^T)^{-1}$. Therefore, $m_{\theta^i}(x)$ is Lipschitz continuous in $\theta^i$. Moreover, following the similar steps as above, we have

$$
\begin{aligned}
\left\| \mathbf{P}_i m_{\theta_1^i}(x) - \mathbf{P}_i m_{\theta_2^i}(x) \right\| &= \left\| \sum_{y \in \mathcal{X}_i} \mathbf{P}_i(x,y) m_{\theta_1^i}(y) - \sum_{y \in \mathcal{X}_i} \mathbf{P}_i(x,y) m_{\theta_2^i}(y) \right\| \\
&= \left\| \sum_{y \in \mathcal{X}_i} \mathbf{P}_i(x,y) \left( m_{\theta_1^i}(y) - m_{\theta_2^i}(y) \right) \right\| \\
&\leq \sum_{y \in \mathcal{X}^i} \mathbf{P}_i(x,y) \left\| m_{\theta_1^i}(y) - m_{\theta_2^i}(y) \right\| \\
&\leq |\mathcal{X}_i| \left\| m_{\theta_1^i}(y) - m_{\theta_2^i}(y) \right\| \\
&\leq |\mathcal{X}_i| (C_i L + 1) \| \theta_1^i - \theta_2^i \|
\end{aligned}
\tag{47}
$$

such that $\mathbf{P}_i m_{\theta^i}(x)$ is also Lipschitz continuous in $\theta^i$, which comletes the proof. $\square$

Now with (42) we can decompose $\nabla F_i(\theta_n^i, X_n^i) - \nabla f_i(\theta_n^i)$ as

$$
\begin{aligned}
\nabla F_i(\theta_n^i, X_n^i) - \nabla f_i(\theta_n^i) &= m_{\theta_n^i}(X_n^i) - \mathbf{P}_i m_{\theta_n^i}(X_n^i) \\
&= \underbrace{m_{\theta_n^i}(X_n^i) - \mathbf{P}_i m_{\theta_n^i}(X_{n-1}^i)}_{e_{n+1}^i} \\
&\quad + \underbrace{\mathbf{P}_i m_{\theta_n^i}(X_{n-1}^i)}_{\nu_n^i} - \underbrace{\mathbf{P}_i m_{\theta_{n+1}^i}(X_n^i)}_{\nu_{n+1}^i} \\
&\quad + \underbrace{\mathbf{P}_i m_{\theta_{n+1}^i}(X_n^i) - \mathbf{P}_i m_{\theta_n^i}(X_n^i)}_{\xi_{n+1}^i}.
\end{aligned}
\tag{48}
$$

Here $\{\gamma_n e_n^i\}$ is a Martingale difference sequence and we need the martingale convergence theorem in Theorem C.3.

**Theorem C.3** (Theorem 6.4.6 [64]). *For an $\mathcal{F}_n$-Martingale $S_n$, set $X_{n-1} = S_n - S_{n-1}$. If for some $1 \leq p \leq 2$,*

$$\sum_{n=1}^{\infty} \mathbb{E}[\|X_{n-1}\|^p | \mathcal{F}_{n-1}] < \infty \quad a.s. \tag{49}$$

*then $S_n$ converges almost surely.* $\qquad\qquad\qquad\qquad\qquad\qquad\qquad\qquad\qquad\qquad\square$

We want to show that $\sum_n \gamma_{n+1}^2 \mathbb{E}[\|e_{n+1}^i\|^2 | \mathcal{F}_n] < \infty$ such that $\sum_n \gamma_n e_n^i$ converges almost surely by Theorem C.3. As we can see in (45), with Lemma C.2 and Assumption 2.4, for a sample path ($\Theta_n$ within a compact set $\Omega$), $\sup_n \|m_{\theta_n^i}(x)\| < \infty$ and $\sup_n \|\mathbf{P}_i m_{\theta_n^i}(x)\| < \infty$ almost surely for all $x \in \mathcal{X}_i$. This ensures that $e_{n+1}^i$ is an $L_2$-bounded martingale difference sequence, i.e., $\sup_n \|e_{n+1}^i\| \leq \sup_n(\|m_{\theta_n^i}(X_{n+1}^i)\| + \|\mathbf{P}_i m_{\theta_n^i}(X_n^i)\|) \leq D_\Omega < \infty$. Together with Assumption 2.3, we get

$$\sum_n \gamma_{n+1}^2 \mathbb{E}[\|e_{n+1}^i\|^2 | \mathcal{F}_n] \leq D_\Omega \sum_n \gamma_{n+1}^2 < \infty \quad a.s. \tag{50}$$

and thus $\sum_n \gamma_n e_n^i$ converges almost surely.

Next, for the term $\nu_n^i$ we have

$$\sum_{k=0}^{p} \gamma_{k+1}(\nu_k^i - \nu_{k+1}^i) = \sum_{k=0}^{p}(\gamma_{k+1} - \gamma_k)\nu_k^i + \gamma_0 \nu_0^i - \gamma_{p+1}\nu_{p+1}^i. \tag{51}$$

As is shown before, for a given sample path, $\|\mathbf{P}_i m_{\theta_n^i}(x)\|$ is bounded almost surely for all $n$ and $x \in \mathcal{X}^i$ such that $\sup_n \|\nu_n^i\| < \infty$ almost surely. Since $\lim_{n\to\infty}(\gamma_{n+1} - \gamma_n) = 0$, we have $\lim_{n\to\infty}(\gamma_{n+1} - \gamma_n)\nu_n^i = 0$. Note that there exists a path-dependent constant $C$ (that bounds $\|\nu_n^i\|$) such that for any $n \geq m$,

$$\left\| \sum_{k=m}^{n}(\gamma_{k+1} - \gamma_k)\nu_k^i \right\| \leq C \sum_{k=m}^{n}(\gamma_k - \gamma_{k+1}) = C(\gamma_m - \gamma_{n+1}) < C\gamma_m. \tag{52}$$

Since $\lim_{n\to\infty} \gamma_n = 0$, there exists a positive integer $M$ such that for all $n \geq m \geq M$, $\gamma_m < \epsilon/C$ and $\|\sum_{k=m}^{n}(\gamma_{k+1} - \gamma_k)\nu_k^i\| < \epsilon$ for every $\epsilon > 0$. Therefore, $\{\sum_{k=0}^{p}(\gamma_{k+1} - \gamma_k)\nu_k^i\}_{p \geq 0}$ is a Cauchy sequence and $\sum_{k=0}^{\infty}(\gamma_{k+1} - \gamma_k)\nu_k^i$ converges by Cauchy convergence criterion. The last term of (51) tends to zero. Therefore, $\sum_{k=0}^{\infty} \gamma_{k+1}(\nu_k^i - \nu_{k+1}^i)$ converges and is finite.

For the last term $\xi_n^i$, Lemma C.2 leads to

$$\frac{1}{N} \sum_{i=1}^{N} \|\xi_{n+1}^i\| \leq \frac{C'}{N} \sum_{i=1}^{N} \|\theta_{n+1}^i - \theta_n^i\| \leq \frac{C'}{\sqrt{N}} \|\Theta_{n+1} - \Theta_n\|. \tag{53}$$

for the Lipschitz constant $C'$ of $\mathbf{P}_i m_{\theta^i}(x)$. However, the relationship between $\theta_n$ and $\theta_{n+1}$ is not obvious in the D-SGD and FL setting due to the update rule (18) with communication matrix $\mathcal{W}_n$, unlike the classical stochastic approximation shown in (38). We come up with the novel decomposition of $\xi_n^i$, which takes the consensus error into account, to solve this issue, i.e.,

$$\begin{aligned} \xi_{n+1}^i = &\left[ \mathbf{P}_i m_{\theta_{n+1}^i}(X_n^i) - \mathbf{P}_i m_{\theta_{n+1}}(X_n^i) \right] + \left[ \mathbf{P}_i m_{\theta_n}(X_n^i) - \mathbf{P}_i m_{\theta_n^i}(X_n^i) \right] \\ &+ \left[ \mathbf{P}_i m_{\theta_{n+1}}(X_n^i) - \mathbf{P}_i m_{\theta_n}(X_n^i) \right]. \end{aligned} \tag{54}$$

Using the Lipschitzness property of $\mathbf{P}_i m_\theta(X)$ in Lemma C.2, we have

$$
\begin{aligned}
\frac{1}{N} \sum_{i=1}^{N} \left\| \xi_{n+1}^i \right\| \leq & \frac{C'}{N} \sum_{i=1}^{N} \left( \left\| \theta_{n+1}^i - \theta_{n+1} \right\| + \left\| \theta_{n+1} - \theta_n \right\| + \left\| \theta_n - \theta_n^i \right\| \right) \\
\leq & \frac{C'}{\sqrt{N}} \left\| \Theta_{n+1} - \frac{1}{N} \left( \mathbf{1}\mathbf{1}^T \otimes \mathbf{I}_d \right) \Theta_{n+1} \right\| + \frac{C'}{\sqrt{N}} \left\| \Theta_n - \frac{1}{N} \left( \mathbf{1}\mathbf{1}^T \otimes \mathbf{I}_d \right) \Theta_n \right\| \\
& + C' \left\| \theta_{n+1} - \theta_n \right\| \\
= & \frac{C'}{\sqrt{N}} \left( \left\| \mathcal{J}_\perp \Theta_{n+1} \right\| + \left\| \mathcal{J}_\perp \Theta_n \right\| \right) + C' \left\| \theta_{n+1} - \theta_n \right\| \\
= & \frac{C'}{\sqrt{N}} \left( \left\| \mathcal{J}_\perp \Theta_{n+1} \right\| + \left\| \mathcal{J}_\perp \Theta_n \right\| \right) + C' \gamma_{n+1} \left\| \frac{1}{N} (\mathbf{1}^T \otimes \mathbf{I}_d) \nabla \mathbf{F}(\Theta_n, \mathbf{X}_n) \right\|.
\end{aligned}
\tag{55}
$$

In Appendix B we have shown $\lim_{n\to\infty} \mathcal{J}_\perp \Theta_n = \mathbf{0}$ almost surely. Moreover, $\left\| \frac{1}{N}(\mathbf{1}^T \otimes \mathbf{I}_d) \nabla \mathbf{F}(\Theta_n, \mathbf{X}_n) \right\|$ is bounded per sample path. Therefore, $\lim_{n\to\infty} \frac{1}{N} \sum_{i=1}^{N} \left\| \xi_{n+1}^i \right\| = 0$ such that $\lim_{n\to\infty} \frac{1}{N} \sum_{i=1}^{N} \xi_{n+1}^i = 0$ almost surely.

To sum up, we decompose (39) into

$$
\theta_{n+1} = \theta_n - \gamma_{n+1} \nabla f(\theta_n) - \gamma_{n+1} r_n - \gamma_{n+1} \frac{1}{N} \sum_{i=1}^{N} \left( e_{n+1}^i + \nu_n^i - \nu_{n+1}^i + \xi_{n+1}^i \right). \tag{56}
$$

Now that $\lim_{p\to\infty} \sum_{n=1}^{p} \frac{1}{N} \sum_{i=1}^{N} \gamma_n e_n^i$ and $\lim_{p\to\infty} \sum_{n=0}^{p} \frac{1}{N} \sum_{i=1}^{N} \gamma_{n+1} (\nu_n^i - \nu_{n+1}^i)$ converge and are finite, $\lim_{n\to\infty} r_n = 0$, $\lim_{n\to\infty} \frac{1}{N} \sum_{i=1}^{N} \xi_n^i = 0$, all the conditions of C3 in Theorem C.1 are satisfied. Additionally, Assumption 2.4 corresponds to C1, Assumption 2.3 meets C2, and C4 is automatically satisfied when we choose the lyapunov function $V(\theta) = f(\theta)$. Therefore, $\limsup_n \inf_{\theta^* \in \mathcal{L}} \left\| \theta_n - \theta^* \right\| = 0$.

## D   Proof of Theorem 3.3

To obtain Theorem 3.3, we need to utilize the existing CLT result for general SA in Theorem D.1 and check all the necessary conditions therein.

**Theorem D.1** (Theorem 2.1 [29]). *Consider the stochastic approximation iteration* (38), *assume*

C1. *Let $\theta^*$ be the root of function $h$, i.e., $h(\theta^*) = 0$, and assume $\lim_{n\to\infty} \theta_n = \theta^*$. Moreover, assume the mean field $h$ is twice continuously differentiable in a neighborhood of $\theta^*$, and the Jacobian $\mathbf{H} \triangleq \nabla h(\theta^*)$ is Hurwitz, i.e., the largest real part of its eigenvalues $B < 0$;*

C2. *The step size $\sum_n \gamma_n = \infty$, $\sum_n \gamma_n^2 < \infty$, and either (i). $\log(\gamma_{n-1}/\gamma_n) = o(\gamma_n)$, or (ii). $\log(\gamma_{n-1}/\gamma_n) \sim \gamma_n/\gamma_\star$ for some $\gamma_\star > 1/2|B|$;*

C3. *$\sup_n \left\| \theta_n^i \right\| < \infty$ almost surely for any $i \in [N]$;*

C4. *(a) $\{e_n\}_{n\geq 0}$ is an $\mathcal{F}_n$-Martingale difference sequence, i.e., $\mathbb{E}[e_n|\mathcal{F}_{n-1}] = 0$, and there exists $\tau > 0$ such that $\sup_{n\geq 0} \mathbb{E}[\|e_n\|^{2+\tau}|\mathcal{F}_{n-1}] < \infty$;*

*(b) $\mathbb{E}[e_{n+1} e_{n+1}^T | \mathcal{F}_n] = \mathbf{U} + \mathbf{D}_n^{(A)} + \mathbf{D}_n^{(B)}$, where $\mathbf{U}$ is a symmetric positive semi-definite matrix and*

$$
\begin{cases}
\mathbf{D}_n^{(A)} \to 0 \quad \text{almost surely,} \\
\lim_n \gamma_n \mathbb{E} \left[ \left\| \sum_{k=1}^{n} \mathbf{D}_k^{(B)} \right\| \right] = 0.
\end{cases}
\tag{57}
$$

C5. *Let $r_n = r_n^{(1)} + r_n^{(2)}$, $r_n$ is $\mathcal{F}_n$-adapted, and*

$$
\begin{cases}
\left\| r_n^{(1)} \right\| = o(\sqrt{\gamma_n}) \quad a.s. \\
\sqrt{\gamma_n} \left\| \sum_{k=1}^{n} r_k^{(2)} \right\| = o(1) \quad a.s.
\end{cases}
\tag{58}
$$

*Then,*

$$\frac{1}{\sqrt{\gamma_n}}(\theta_n - \theta^*) \xrightarrow[n\to\infty]{dist.} \mathcal{N}(0, \mathbf{V}), \tag{59}$$

*where*

$$\begin{cases} \mathbf{V}\mathbf{H}^T + \mathbf{H}\mathbf{V} = -\mathbf{U} & \text{in case C2 (i),} \\ \mathbf{V}(\mathbf{I}_d + 2\gamma_\star\mathbf{H}^T) + (\mathbf{I}_d + 2\gamma_\star\mathbf{H})\mathbf{V} = -2\gamma_\star\mathbf{U} & \text{in case C2 (ii).} \end{cases} \tag{60}$$

$\square$

Note that the matrix $\mathbf{U}$ in the condition C4(b) of Theorem D.1 was assumed to be positive definite in the original Theorem 2.1 [29]. It was only to ensure that the solution $\mathbf{V}$ to the Lyapunov equation (60) is positive definite, which was only used for the stability of the related autonomous linear ODE (e.g., Theorem 3.16 [15] or Theorem 2.2.3 [36]). However, in this paper, we do not need strict positive definite matrix $\mathbf{V}$. Therefore, we extend $\mathbf{U}$ to be positive semi-definite such that $\mathbf{V}$ is also positive semi-definite (see Lemma D.2 for the closed form of matrix $\mathbf{V}$). Such kind of extension does not change any of the proof steps in [29].

### D.1 Discussion about C1-C3

Our Assumption 2.1 corresponds to C1 by letting function $h(\theta) = -\nabla f(\theta)$ therein. We can also let $\gamma_\star$ in Theorem 3.3 large enough to satisfy C2. The typical form of step size, also indicated in [29], is polynomial step size $\gamma_n \sim \gamma_\star/n^a$ for $a \in (0.5, 1]$. Note that $a \in (0.5, 1)$ satisfies C2 (i) and $a = 1$ satisfies C2 (ii). Assumption 2.4 corresponds to C3.[4]

### D.2 Analysis of C4

To check condition C4, we need to analyze the Martingale difference sequence $\{e_n^i\}$. Recall $e_{n+1}^i = m_{\theta_n^i}(X_n^i) - \mathbf{P}_i m_{\theta_n^i}(X_{n-1}^i)$ such that there exists a constant $C$,

$$\begin{aligned} \mathbb{E}\left[\left\|e_{n+1}^i\right\|^{2+\tau}\big|\mathcal{F}_n\right] &\leq C\mathbb{E}\left[\left\|m_{\theta_n^i}(X_n^i)\right\|^{2+\tau} + \left\|\mathbf{P}_i m_{\theta_n^i}(X_{n-1}^i)\right\|^{2+\tau}\Big|\mathcal{F}_n\right] \\ &= C\sum_{Y\in\mathcal{X}^i}\mathbf{P}_i(X_{n-1}^i, Y)\left\|m_{\theta_n^i}(Y)\right\|^{2+\tau} + C\left\|\mathbf{P}_i m_{\theta_n^i}(X_{n-1}^i)\right\|^{2+\tau}. \end{aligned} \tag{61}$$

Since $\left\|m_{\theta_n^i}(Y)\right\| < \infty$ almost surely by Assumption 2.4 and $\mathcal{X}^i$ is a finite state space, at all time $n$, we have

$$\sum_{Y\in\mathcal{X}^i}\mathbf{P}_i(X_{n-1}^i, Y)\left\|m_{\theta_n^i}(Y)\right\|^{2+\tau} < \infty \quad a.s. \tag{62}$$

and there exists another constant $C'$ such that by definition of $\mathbf{P}_i m_{\theta_n^i}(X_{n-1}^i)$, we have

$$\left\|\mathbf{P}_i m_{\theta_n^i}(X_{n-1}^i)\right\|^{2+\tau} \leq C'\sum_{Y\in\mathcal{X}^i}\mathbf{P}_i(X_{n-1}^i, Y)\left\|m_{\theta_n^i}(Y)\right\|^{2+\tau} < \infty \quad a.s. \tag{63}$$

Therefore, $\mathbb{E}[\|e_{n+1}^i\|^{2+\tau}|\mathcal{F}_n] < \infty$ a.s. for all $n$ and C4.(a) is satisfied.

We now turn to C4.(b). Note that for any $i \neq j$, we have $\mathbb{E}[e_{n+1}^i(e_{n+1}^j)^T|\mathcal{F}_n] = \mathbb{E}[e_{n+1}^i|\mathcal{F}_n] \cdot \mathbb{E}[(e_{n+1}^j)^T|\mathcal{F}_n] = 0$ due to the independence between agent $i$ and $j$, and $\mathbb{E}[e_{n+1}^i|\mathcal{F}_n] = 0$. Then, we have

$$\mathbb{E}\left[\left(\frac{1}{N}\sum_{i=1}^N e_{n+1}^i\right)\left(\frac{1}{N}\sum_{i=1}^N e_{n+1}^i\right)^T\Bigg|\mathcal{F}_n\right] = \frac{1}{N^2}\sum_{i=1}^N\mathbb{E}\left[e_{n+1}^i(e_{n+1}^i)^T\big|\mathcal{F}_n\right]. \tag{64}$$

The analysis of $\mathbb{E}[e_{n+1}^i(e_{n+1}^i)^T|\mathcal{F}_n]$ is inspired by Section 4 [29] and Section 4.3.3 [22], where they constructed another Poisson equation to further decompose the noise terms therein.[5] Here, expanding

---

[4]Theorem D.1 is slightly modified in terms of condition C3, which is mentioned as a special case in Section 2.2 [29]. For the sake of mathematical simplicity, we stick to condition C3 in the proof.

[5]However, we note that [29, 22] considered the Lipschitz continuity of function $F_{\theta^i}^i(x)$ defined in (68) as an assumption instead of a conclusion, where we give a detailed proof for this. We also obtain matrix $\mathbf{U}_i$ in an explicit form, which coincides with the definition of *asymptotic covariance matrix* and was not simplified in [29]. The discussion on the improvement of $\mathbf{U}_i$ is outlined in Section 3.2, which was not the focus of [29, 22] and was not covered therein.

$\mathbb{E}[e_{n+1}^i(e_{n+1}^i)^T|\mathcal{F}_n]$ gives

$$
\begin{aligned}
\mathbb{E}\left[e_{n+1}^i(e_{n+1}^i)^T\middle|\mathcal{F}_n\right] =&\mathbb{E}[m_{\theta_n^i}(X_n^i)m_{\theta_n^i}(X_n^i)^T|\mathcal{F}_n] + \mathbf{P}_i m_{\theta_n^i}(X_{n-1}^i)\left(\mathbf{P}_i m_{\theta_n^i}(X_{n-1}^i)\right)^T\\
&-\mathbb{E}[m_{\theta_n^i}(X_n^i)|\mathcal{F}_n]\left(\mathbf{P}_i m_{\theta_n^i}(X_{n-1}^i)\right)^T - \mathbf{P}_i m_{\theta_n^i}(X_{n-1}^i)\mathbb{E}[m_{\theta_n^i}(X_n^i)^T|\mathcal{F}_n]\\
=&\sum_{y\in\mathcal{X}_i}\mathbf{P}_i(X_{n-1},y)m_{\theta_n^i}(y)m_{\theta_n^i}(y)^T - \mathbf{P}_i m_{\theta_n^i}(X_{n-1}^i)\left(\mathbf{P}_i m_{\theta_n^i}(X_{n-1}^i)\right)^T.
\end{aligned}
\tag{65}
$$

Denote by

$$
G_i(\theta^i,x) \triangleq \sum_{y\in\mathcal{X}_i}\mathbf{P}_i(x,y)m_{\theta^i}(y)m_{\theta^i}(y)^T - \mathbf{P}_i m_{\theta^i}(x)\left(\mathbf{P}_i m_{\theta^i}(x)\right)^T,
\tag{66}
$$

and let its expectation w.r.t the stationary distribution $\boldsymbol{\pi}_i$ be $g_i(\theta^i) \triangleq \mathbb{E}_{x\sim\boldsymbol{\pi}_i}[G_i(\theta^i,x)]$, we can construct another Poisson equation, i.e.,

$$
\begin{aligned}
&\mathbb{E}\left[e_{n+1}^i(e_{n+1}^i)^T\middle|\mathcal{F}_n\right] - \sum_{X_n^i\in\mathcal{X}_i}\boldsymbol{\pi}(X_n^i)\mathbb{E}\left[e_{n+1}^i(e_{n+1}^i)^T\middle|\mathcal{F}_n\right]\\
=&G_i(\theta_n^i,X_{n-1}^i) - g_i(\theta_n^i)\\
=&\varphi_{\theta_n^i}^i(X_{n-1}^i) - \mathbf{P}_i\varphi_{\theta_n^i}^i(X_{n-1}^i),
\end{aligned}
\tag{67}
$$

for some matrix-valued function $\varphi^i:\mathbb{R}^d\times\mathcal{X}_i\to\mathbb{R}^{d\times d}$. Following the similar steps shown in (42) - (45), we can obtain the closed-form expression

$$
\varphi_{\theta^i}^i(x) = \sum_{y\in\mathcal{X}_i}\left(\mathbf{I} - \mathbf{P}_i + \mathbf{1}(\boldsymbol{\pi}_i)^T\right)^{-1}(x,y)G_i(\theta^i,x) - g_i(\theta^i).
\tag{68}
$$

Then, we can decompose (65) into

$$
G_i(\theta_n^i,X_{n-1}^i) = \underbrace{g_i(\theta^*)}_{\mathbf{U}_i} + \underbrace{g_i(\theta_n^i) - g_i(\theta^*)}_{\mathbf{D}_{i,n}^{(1)}} + \underbrace{\varphi_{\theta_n^i}^i(X_n^i) - \mathbf{P}_i\varphi_{\theta_n^i}^i(X_{n-1}^i)}_{\mathbf{D}_{i,n}^{(2,a)}} + \underbrace{\varphi_{\theta_n^i}^i(X_{n-1}^i) - \varphi_{\theta_n^i}^i(X_n^i)}_{\mathbf{D}_{i,n}^{(2,b)}}.
\tag{69}
$$

Let $\mathbf{U} \triangleq \frac{1}{N^2}\sum_{i=1}^N\mathbf{U}_i$, $\mathbf{D}_n^{(1)} \triangleq \frac{1}{N^2}\sum_{i=1}^N\mathbf{D}_{1,n}^{(1)}$, $\mathbf{D}_n^{(2,a)} \triangleq \frac{1}{N^2}\sum_{i=1}^N\mathbf{D}_{i,n}^{(2,a)}$, and $\mathbf{D}_n^{(2,b)} \triangleq \frac{1}{N^2}\sum_{i=1}^N\mathbf{D}_{i,n}^{(2,b)}$, we want to prove that $\mathbf{D}_n^{(1)}$ satisfies the first condition in C4, and $\mathbf{D}_n^{(2,a)},\mathbf{D}_n^{(2,b)}$ meet the second condition in C4.

We now show that for all $i$, $G_i(\theta^i,x)$ is Lipschitz continuous in $\theta^i\in\Omega$ for some compact subset $\Omega\subset\mathbb{R}^d$. For any $x\in\mathcal{X}_i$ and $\theta_1^i,\theta_2^i\in\Omega$, we can get

$$
\begin{aligned}
&\|m_{\theta_1^i}(x)m_{\theta_1^i}(x)^T - m_{\theta_2^i}(x)m_{\theta_2^i}(x)^T\|\\
=&\|m_{\theta_1^i}(x)(m_{\theta_1^i}(x) - m_{\theta_2^i}(x))^T - (m_{\theta_1^i}(x) - m_{\theta_2^i}(x))m_{\theta_2^i}(x)^T\|\\
\leq&\|m_{\theta_1^i}(x) - m_{\theta_2^i}(x)\|(\|m_{\theta_1^i}(x)\| + \|m_{\theta_2^i}(x)\|)\\
\leq&C\|\theta_1^i - \theta_2^i\|,
\end{aligned}
\tag{70}
$$

for some constant $C$, where the last inequality comes from $\|m_{\theta_1^i}(x)\| < \infty$ since $\theta_1^i\in\Omega$ and the Lipschitz continuous function $m_{\theta^i}(x)$. Similarly, we can get $\|\mathbf{P}_i m_{\theta_1^i}(x) - \mathbf{P}_i m_{\theta_2^i}(x)\| \leq C\|\theta_1^i - \theta_2^i\|$. Therefore, $G_i(\theta^i,x)$ and $g_i(\theta^i)$ are Lipschitz continuous in $\theta^i\in\Omega$ for any $x\in\mathcal{X}_i$.

For the sequence $\{\mathbf{D}_{i,n}^{(1)}\}_{n\geq0}$, by applying Theorem 3.2 and conditioned on $\lim_{n\to\infty}\theta_n = \theta^*$ for an optimal point $\theta^*\in\mathcal{L}$, we have $\lim_{n\to\infty}\|g_i(\theta_n^i) - g_i(\theta^*)\| \leq \lim_{n\to\infty}C\|\theta_n^i - \theta^*\| = 0$. This implies $\mathbf{D}_{i,n}^{(1)}\to 0$ for every $i\in[N]$ and thus $\mathbf{D}_n^{(1)}\to 0$ as $n\to\infty$ almost surely, which satisfies the first condition in (57).

For the Martingale difference sequence $\{\mathbf{D}_{i,n}^{(2,a)}\}_{n\geq0}$, we use Burkholder inequality (e.g., Theorem 2.10 [33], [21]) such that for $p\geq 1$ and some constant $C_p$,

$$
\mathbb{E}\left[\left\|\sum_{i=1}^n\mathbf{D}_{i,n}^{(2,a)}\right\|^p\right] \leq C_p\mathbb{E}\left[\left(\sum_{i=1}^n\left\|\mathbf{D}_{i,n}^{(2,a)}\right\|^2\right)^{p/2}\right].
\tag{71}
$$

By the definition (66) and Assumption 2.4, for a sample path, $\sup_n \|G_i(\theta_n^i, x)\| < \infty$ for any $x \in \mathcal{X}_i$, as well as $\sup_n \|g_i(\theta_n^i)\| < \infty$, which leads to $\sup_n \|\varphi_{\theta_n^i}^i(x)\| < \infty$ for any $x \in \mathcal{X}_i$ because of (68).

Then, we have $\sup_n \|\mathbf{D}_{i,n}^{(2,a)}\| \leq C < \infty$ for the path-dependent constant $C$. Taking $p = 1$ and we have

$$\lim_{n\to\infty} \gamma_n C_p \sqrt{\sum_{i=1}^{n} \left\|\mathbf{D}_{i,n}^{(2,a)}\right\|^2} \leq \lim_{n\to\infty} C_p C \gamma_n \sqrt{n} = 0 \quad a.s. \tag{72}$$

Thus, Lebesgue dominated convergence theorem gives

$$\lim_{n\to\infty} \gamma_n C_p \mathbb{E}\left[\sqrt{\sum_{i=1}^{n} \|\mathbf{D}_{i,n}^{(2,a)}\|^2}\right] = \mathbb{E}\left[\lim_{n\to\infty} \gamma_n C_p \sqrt{\sum_{i=1}^{n} \|\mathbf{D}_{i,n}^{(2,a)}\|^2}\right] = 0$$

and we have $\lim_{n\to\infty} \gamma_n \mathbb{E}[\|\sum_{i=1}^{n} \mathbf{D}_{i,n}^{(2,a)}\|] = 0$.

For the sequence $\{\mathbf{D}_{i,n}^{(2,b)}\}_{n\geq 0}$, we have

$$\sum_{k=1}^{n} \mathbf{D}_{i,k}^{(2,b)} = \sum_{k=1}^{n} \left(\varphi_{\theta_k^i}^i(X_{k-1}^i) - \varphi_{\theta_{k-1}^i}^i(X_{k-1}^i)\right) + \varphi_{\theta_0^i}^i(X_0^i) - \varphi_{\theta_n^i}^i(X_n^i)$$

$$= \sum_{k=1}^{n} \left(\varphi_{\theta_k^i}^i(X_{k-1}^i) - \varphi_{\theta_k}^i(X_{k-1}^i) + \varphi_{\theta_k}^i(X_{k-1}^i) - \varphi_{\theta_{k-1}}^i(X_{k-1}^i) + \varphi_{\theta_{k-1}}^i(X_{k-1}^i) - \varphi_{\theta_{k-1}^i}^i(X_{k-1}^i)\right)$$

$$+ \varphi_{\theta_0^i}^i(X_0^i) - \varphi_{\theta_n^i}^i(X_n^i). \tag{73}$$

Since $G_i(\theta^i, x)$ and $g_i(\theta^i)$ are Lipschitz continuous in $\theta^i \in \Omega$, $\varphi_{\theta^i}^i(x)$ is also Lipschitz continuous in $\theta^i \in \Omega$ and is bounded. We have

$$\left\|\sum_{k=1}^{n} \mathbf{D}_{i,k}^{(2,b)}\right\| \leq \left\|\sum_{k=1}^{n} \varphi_{\theta_k^i}^i(X_{k-1}^i) - \varphi_{\theta_{k-1}^i}^i(X_{k-1}^i)\right\| + \left\|\varphi_{\theta_0^i}^i(X_0^i)\right\| + \left\|\varphi_{\theta_n^i}^i(X_n^i)\right\|$$

$$\leq \left\|\sum_{k=1}^{n} \varphi_{\theta_k^i}^i(X_{k-1}^i) - \varphi_{\theta_{k-1}^i}^i(X_{k-1}^i)\right\| + D_1 \tag{74}$$

$$\leq \sum_{k=1}^{n} D_2 D_\Omega \gamma_k + D_1$$

where $\|\varphi_{\theta_0^i}^i(X_0^i)\| + \|\varphi_{\theta_n^i}^i(X_n^i)\| \leq D_1$ for a given sample path, $D_2$ is the Lipschitz constant of $\varphi_{\theta^i}^i(x)$, and $\|\nabla F_i(x^i, X^i)\| \leq D_\Omega$ for any $x^i \in \Omega$ and $X^i \in \mathcal{X}^i$. Then,

$$\gamma_n \left\|\sum_{k=1}^{n} \mathbf{D}_{i,k}^{(2,b)}\right\| \leq D_2 D_\Omega \gamma_n \sum_{k=1}^{n} \gamma_k + \gamma_n D_1 \to 0 \quad \text{as } n \to \infty \tag{75}$$

because $\gamma_n \sum_{k=1}^{n} \gamma_k = O(n^{1-2a})$ by assumption 2.3. Therefore, the second condition of C4 is satisfied.

## D.3 Analysis of C5

We now analyze condition C5. The decreasing rate of each term in (56) has been proved in Appendix C. Specifically, by assumption 2.4, there exists a compact subset for a given sample path, and

- we have shown that $\left\|r_n^{(A)}\right\| = O(\eta_n)$ a.s., which implies $\left\|r_n^{(A)}\right\| = o(\sqrt{\gamma_n})$ a.s.

- For $\frac{1}{N} \sum_{i=1}^{N} \xi_n^i$, in the case of increasing communication interval, $\frac{1}{N} \sum_{i=1}^{N} \xi_n^i = O(\gamma_n + \eta_n)$, by Assumption 2.3-ii), we know $(\gamma_n + \eta_n)/\sqrt{\gamma_n} = \sqrt{\gamma_n} + \sqrt{\gamma_n} K_{\tau_n}^{L+1} = o(1)$ such that $\|\frac{1}{N} \sum_{i=1}^{N} \xi_n^i\| = o(\sqrt{\gamma_n})$ almost surely. On the other hand, in the case of bounded communication interval, $\frac{1}{N} \sum_{i=1}^{N} \xi_n^i = O(\gamma_n)$ such that $\|\frac{1}{N} \sum_{i=1}^{N} \xi_n^i\| = o(\sqrt{\gamma_n})$ a.s.

- Since $\sup_n \|\nu_n^i\| < \infty$ almost surely, we have $\sup_p \|\frac{1}{N}\sum_{i=1}^{N}\sum_{k=0}^{p}(\nu_k^i - \nu_{k+1}^i)\| = \sup_p \|\frac{1}{N}\sum_{i=1}^{N}(\nu_0^i - \nu_{p+1}^i)\| < \infty$ almost surely. Then, $\sqrt{\gamma_p}\|\frac{1}{N}\sum_{i=1}^{N}\sum_{k=0}^{p}(\nu_k^i - \nu_{k+1}^i)\| = O(\sqrt{\gamma_p})$ leads to $\sqrt{\gamma_p}\|\frac{1}{N}\sum_{i=1}^{N}\sum_{k=0}^{p}(\nu_k^i - \nu_{k+1}^i)\| = o(1)$ a.s.

Let $r_n^{(1)} \triangleq r_n^{(A)} + \frac{1}{N}\sum_{i=1}^{N}\xi_n^i$ and $r_n^{(2)} \triangleq \frac{1}{N}\sum_{i=1}^{N}(\nu_k^i - \nu_{k+1}^i)$. From above, we can see that C5 in Theorem D.1 is satisfied and we show that all the conditions in Theorem D.1 have been satisfied.

### D.4   CLosed Form of Limitimg Covariance Matrix

Lastly, we need to analyze the closed-form expression of $\mathbf{U}$ as in C4 (b) of Theorem D.1. Recall that $\mathbf{U} = \frac{1}{N^2}\sum_{i=1}^{N}\mathbf{U}_i$ and $\mathbf{U}_i = g_i(\theta^*)$ in (69). We now give the exact form of function $g_i(\theta^*)$ as follows:

$$
\begin{aligned}
g_i(\theta^*) &= \sum_{x\in\mathcal{X}_i}\boldsymbol{\pi}_i(x)\left[m_{\theta^*}(x)m_{\theta^*}(x)^T - \left(\sum_{y\in\mathcal{X}_i}\mathbf{P}_i(x,y)m_{\theta^*}(y)\right)\left(\sum_{y\in\mathcal{X}_i}\mathbf{P}_i(x,y)m_{\theta^*}(y)\right)^T\right]\\
&= \mathbb{E}\left[\left(\sum_{s=0}^{\infty}[\nabla F_i(\theta^*,X_s) - \nabla f_i(\theta^*)]\right)\left(\sum_{s=0}^{\infty}[\nabla F_i(\theta^*,X_s) - \nabla f_i(\theta^*)]\right)^T\right]\\
&\quad - \mathbb{E}\left[\left(\sum_{s=1}^{\infty}[\nabla F_i(\theta^*,X_s) - \nabla f_i(\theta^*)]\right)\left(\sum_{s=1}^{\infty}[\nabla F_i(\theta^*,X_s) - \nabla f_i(\theta^*)]\right)^T\right]\\
&= \mathbb{E}\left[\left(\nabla F_i(\theta^*,X_0^i) - \nabla f_i(\theta^*)\right)\left(\nabla F_i(\theta^*,X_0^i) - \nabla f_i(\theta^*)\right)^T\right]\\
&\quad + \mathbb{E}\left[\left(\nabla F_i(\theta^*,X_0^i) - \nabla f_i(\theta^*)\right)\left(\sum_{s=1}^{\infty}[\nabla F_i(\theta^*,X_s) - \nabla f_i(\theta^*)]\right)^T\right]\\
&\quad + \mathbb{E}\left[\left(\sum_{s=1}^{\infty}[\nabla F_i(\theta^*,X_s) - \nabla f_i(\theta^*)]\right)\left(\nabla F_i(\theta^*,X_0^i) - \nabla f_i(\theta^*)\right)^T\right]\\
&= \mathrm{Cov}(\nabla F_i(\theta^*,X_0),\nabla F_i(\theta^*,X_0))\\
&\quad + \sum_{s=1}^{\infty}\left[\mathrm{Cov}(\nabla F_i(\theta^*,X_0),\nabla F_i(\theta^*,X_s)) + \mathrm{Cov}(\nabla F_i(\theta^*,X_s),\nabla F_i(\theta^*,X_0))\right],\\
&= \boldsymbol{\Sigma}(\nabla F(\theta^*,\cdot)).
\end{aligned}
$$

(76)

where the second equality comes from the recursive form of $m_{\theta^i}(x)$ in (45), and that the process $\{X_n\}_{n\geq 0}$ is in its stationary regime, i.e., $X_0 \sim \boldsymbol{\pi}_i$ from the beginning. The last equality comes from rewriting $\mathrm{Cov}(\nabla F_i(\theta^*,X_i),\nabla F_i(\theta^*,X_j))$ in a matrix form. Note that $g_i(\theta^*)$ is exactly the asymptotic covariance matrix of the underlying Markov chain $\{X_n^i\}_{n\geq 0}$ associated with the test function $\nabla F_i(\theta^*,\cdot)$. By utilizing the following lemma, we can obtain the explicit form of $\mathbf{V}$ as defined in (60).

**Lemma D.2** (Lemma D.2.2 [41]). *If all the eigenvalues of matrix $\mathbf{M}$ have negative real part, then for every positive semi-definite matrix $\mathbf{U}$ there exists a unique positive semi-definite matrix $\mathbf{V}$ satisfying $\mathbf{U} + \mathbf{M}\mathbf{V} + \mathbf{V}\mathbf{M}^T = \mathbf{0}$. The explicit solution $\mathbf{V}$ is given as*

$$
\mathbf{V} = \int_0^{\infty} e^{\mathbf{M}t}\mathbf{U}e^{(\mathbf{M}^T)t}dt. \tag{77}
$$

### D.5   CLT of Polyak-Ruppert Averaging

We now consider the CLT result of Polyak-Ruppert averaging $\bar{\theta}_n = \frac{1}{n}\sum_{k=0}^{n-1}\theta_k$. The steps follow similar way by verifying that the conditions in the related CLT of Polyak-Ruppert averaging for the stochastic approximation are satisfied. The additional assumption is given below.

C6. For the sequence $\{r_n\}$ in (38), $n^{-1/2} \sum_{k=0}^{n} r_k^{(1)} \to 0$ with probability 1.

Then, the CLT of Polyak-Ruppert averaging is as follows.

**Theorem D.3** (Theorem 3.2 of [29]). *Consider the iteration (38), assume C1, C3, C4, C5 in Theorem D.1 are satisfied. Moreover, assume C6 is satisfied. Then, with step size $\gamma_n \sim \gamma_\star/n^a$ for $a \in (0.5, 1)$, we have*

$$\sqrt{n}(\bar{\theta}_n - \theta^*) \xrightarrow[n\to\infty]{dist.} \mathcal{N}(0, \mathbf{V}'), \tag{78}$$

*where $\mathbf{V}' = \mathbf{H}^{-1}\mathbf{U}\mathbf{H}^{-T}$.*

Discussion about C1 and C3 can be found in Section D.1. Condition C4 has been analyzed in Section D.2 and condition C5 has been examined in Section D.3. The only condition left to analyze is C6, which is based on the results obtained in Section D.3. In view of (56), $r_n^{(1)} = r_n^{(A)} + \frac{1}{N} \sum_{i=1}^{N} \xi_{n+1}^i$, so C6 is equivalent to

$$n^{-1/2} \sum_{k=1}^{n} \left[ r_k^{(A)} + \frac{1}{N} \sum_{i=1}^{N} \left( \xi_{k+1}^i \right) \right] \to 0 \quad w.p.1. \tag{79}$$

In Section D.3, we have shown that $\left\| r_n^{(A)} \right\| = O(\eta_n)$, $\frac{1}{N} \sum_{i=1}^{N} \xi_n^i = O(\gamma_n)$. Note that by Assumption 2.3, we consider bounded communication interval for step size $\gamma_n \sim \gamma_\star/n^a$ for $a \in (0.5, 1)$, and hence, $\eta_n = O(\gamma_n)$ such that $\left\| r_n^{(A)} \right\| = O(\gamma_n)$. We then know that

$$\sum_{k=1}^{n} \left\| r_n^{(A)} \right\| = O(n^{1-a}), \quad \sum_{k=1}^{n} \left\| \frac{1}{N} \sum_{i=1}^{N} \xi_n^i \right\| = O(n^{1-a}), \tag{80}$$

such that

$$n^{-1/2} \sum_{k=1}^{n} \left\| r_k^{(A)} + \frac{1}{N} \sum_{i=1}^{N} \left( \xi_{k+1}^i \right) \right\| = O(n^{1/2-a}) = o(1), \tag{81}$$

which proved (79) and C6 is verified. Therefore, Theorem D.3 is proved under our Assumptions 2.1 - 2.5.

# E  Discussion on the comparison of Theorem 3.3 to the CLT result in [51]

As a byproduct of our Theorem 3.3, we have the following corollary.

**Corollary E.1.** *Under Assumptions 2.1 - 2.5, for the sub-sequence $\{n_l\}_{l \geq 0}$ where $K_l = K$ for all $l$, we have*

$$\frac{1}{\sqrt{n_l}} \sum_{k=1}^{l} (\bar{\theta}_{n_k} - \theta^*) \xrightarrow[l\to\infty]{dist.} \mathcal{N}(0, \mathbf{V}') \tag{82}$$

*Proof.* Since $K_l = K$ for all $l$, we have $n_l = Kl$. There is an existing result showing the CLT result of the partial sum of a sub-sequence (after normalization) has the same normal distribution as the partial sum of the original sequence.

**Theorem E.2** (Theorem 14.4 of [8]). *Given a sequence of random variable $\theta_1, \theta_2, \cdots$ with partial sum $S_n \triangleq \sum_{k=1}^{n} \theta_k$ such that $\frac{1}{\sqrt{n}} S_n \xrightarrow[n\to\infty]{dist.} \mathcal{N}(0, \mathbf{V})$. Let $n_l$ be some positive random variable taking integer value such that $\theta_{n_l}$ is on the same space as $\theta_n$. In addition, for some sequence $\{b_l\}_{l\geq 0}$ going to infinity, $n_l/b_l \to c$ for a positive constant $c$. Then, $\frac{1}{\sqrt{n_l}} S_{n_l} \xrightarrow[l\to\infty]{dist.} \mathcal{N}(0, \mathbf{V})$.*

From Theorem E.2 and our Theorem 3.3, we have $\frac{1}{\sqrt{n_l}} \sum_{k=1}^{l} (\bar{\theta}_{n_k} - \theta^*) \xrightarrow[l\to\infty]{dist.} \mathcal{N}(0, \mathbf{V}')$. $\qquad \square$

Recently, [51] studied the CLT result under the LSGD-FP algorithm with *i.i.d* sampling (with slightly different setting of the step size). We are able recover Theorem 3.1 of [51] under the constant communication interval while adjusting their step size to make a fair comparison. We state their algorithm below for self-contained purpose. During each communication interval $n \in (n_l, n_{l+1}]$,

$$\theta_{n+1}^i = \begin{cases} \theta_n^i - \gamma_l \nabla F_i(\theta_n^i, X_n^i) & \text{if } n \in (n_l, n_{l+1}), \\ \frac{1}{N} \sum_{i=1}^N (\theta_n^i - \gamma_l \nabla F_i(\theta_n^i, X_n^i)) & \text{if } n = n_{l+1}. \end{cases} \tag{83}$$

The CLT result associated with (83) is given below.

**Theorem E.3** (Theorem 3.1 of [51]). *Under LSGD-FP algorithm with i.i.d. sampling, we have*

$$\frac{\sqrt{n_l}}{l} \sum_{k=1}^l \left(\bar{\theta}_{n_k} - \theta^*\right) \xrightarrow[l\to\infty]{dist.} \mathcal{N}(0, \nu \mathbf{V}'), \tag{84}$$

*where* $\nu \triangleq \lim_{l\to\infty} \frac{1}{l^2} (\sum_{k=1}^l K_l)(\sum_{k=1}^l K_l^{-1})$.

Note that $\nu = 1$ for constant $K$. We can rewrite (84) as

$$\frac{\sqrt{n_l}}{l} \sum_{k=1}^l \left(\bar{\theta}_{n_k} - \theta^*\right) = \frac{\sqrt{n_l}}{\sqrt{l}} \frac{1}{\sqrt{l}} \sum_{k=1}^l (\bar{\theta}_{n_k} - \theta^*) = \sqrt{K} \frac{1}{\sqrt{l}} \sum_{k=1}^l (\bar{\theta}_{n_k} - \theta^*) \tag{85}$$

such that

$$\frac{1}{\sqrt{l}} \sum_{k=1}^l (\bar{\theta}_{n_k} - \theta^*) \xrightarrow[l\to\infty]{dist.} \mathcal{N}(0, \frac{1}{K} \mathbf{V}'). \tag{86}$$

Note that the step size in (83) keeps unchanged during each communication interval, while ours in (1) keeps decreasing even in the same communication interval. This makes our step size decreasing faster than theirs. To make a fair comparison, we only choose a sub-sequence $\{n_{Kl}\}_{l\geq 0}$ in (86) such that it is 'equivalent' to see that our step sizes become the same at each aggregation step. In this case, we again use Theorem E.2 to obtain

$$\frac{1}{\sqrt{Kl}} \sum_{s=1}^l (\bar{\theta}_{n_{Ks}} - \theta^*) \xrightarrow[l\to\infty]{dist.} \mathcal{N}(0, \frac{1}{K} \mathbf{V}'), \tag{87}$$

such that

$$\frac{1}{\sqrt{l}} \sum_{s=1}^l (\bar{\theta}_{n_{Ks}} - \theta^*) = \sqrt{K} \frac{1}{\sqrt{Kl}} \sum_{s=1}^l (\bar{\theta}_{n_{Ks}} - \theta^*) \xrightarrow[l\to\infty]{dist.} \mathcal{N}(0, \mathbf{V}'). \tag{88}$$

Therefore, our Corollary E.1 also recovers Theorem 3.1 of [51] under the constant communication interval $K$, but with more general communication patterns and *Markovian* sampling.

# F  Discussion on Communication Patterns

## F.1  Examples of Communication Matrix W

**Metropolis Hasting Algorithm:** In the decentralized learning such as D-SGD, HLSGD and DFL, $\mathbf{W}$ at the aggregation step can be generated locally using the Metropolis Hasting algorithm based on the underlying communication topology, and is deterministic [62, 43, 80]. Specifically, each agent $i$ exchanges its degree $d_i$ with its neighbors $j \in N(i)$, forming the weight $\mathbf{W}(i,j) = \min\{1/d_i, 1/d_j\}$ for $j \in N(i)$ and $\mathbf{W}(i,i) = 1 - \sum_{j \neq N(i)} \mathbf{W}(i,j)$. In this case, $\mathbf{W}$ is doubly stochastic and symmetric. By Perron-Frobenius theorem, its SLEM $\lambda_2(\mathbf{W}) < 1$. Then, $\|\mathbf{W}^T \mathbf{W} - \mathbf{J}\| = \|\mathbf{W}^2 - \mathbf{J}\| = \lambda_2^2(\mathbf{W}) < 1$, which satisfies Assumption 2.5-ii). It is worth noting that this algorithm is robust to time-varying communication topologies.

**Client Sampling in FL:** For LSGD-FP studied in [67, 76, 42], $\mathbf{W} = \mathbf{1}\mathbf{1}^T/N$ trivially satisfies Assumption 2.5-ii). For LSGD-PP on the other hand, only a small fraction of agents participate in each aggregation step for consensus [50, 30]. Denote by $\mathcal{S}$ a randomly selected set of agents (without replacement) of fixed size $|\mathcal{S}| \in \{1, 2, \cdots, N\}$ at time $n$ and $\mathbf{W}_{\mathcal{S}}$ plays a role of aggregating $\theta_n^i$

for agent $i \in \mathcal{S}$. Additionally, the central server needs to broadcast updated parameter $\theta_{n+1}$ to the newly selected set $\mathcal{S}'$ with the same size, which results in a bijective mapping $\sigma$ (for $\mathcal{S} \to \mathcal{S}'$ and $[N]/\mathcal{S} \to [N]/\mathcal{S}'$) and a corresponding permutation matrix $\mathbf{T}_{\mathcal{S} \to \mathcal{S}'}$. Then, the communication matrix becomes $\mathbf{W} = \mathbf{T}_{\mathcal{S} \to \mathcal{S}'} \mathbf{W}_{\mathcal{S}}$.[6] Specifically, $\mathbf{T}_{\mathcal{S} \to \mathcal{S}'}(i, j) = 1$ if $j = \sigma(i)$ and $\mathbf{T}_{\mathcal{S} \to \mathcal{S}'}(i, j) = 0$ otherwise. Besides, $\mathbf{W}_{\mathcal{S}}(i, j) = 1/|\mathcal{S}|$ for $i, j \in \mathcal{S}$, $\mathbf{W}_{\mathcal{S}}(i, i) = 1$ for $i \notin \mathcal{S}$, and $\mathbf{W}_{\mathcal{S}}(i, j) = 0$ otherwise. Note that $\mathbf{W}_{\mathcal{S}}$ is now a random matrix, since $\mathcal{S}$ is a randomly chosen subset of size $|\mathcal{S}|$. Clearly, for each choice of $\mathcal{S}$, $\mathbf{W}_{\mathcal{S}}$ is doubly stochastic, symmetric and $\mathbf{W}_{\mathcal{S}}^2 = \mathbf{W}_{\mathcal{S}}$. Taking the expectation of $\mathbf{W}_{\mathcal{S}}$ w.r.t the randomly selected set $\mathcal{S}$ gives $\mathbb{E}_{\mathcal{S}}[\mathbf{W}_{\mathcal{S}}](i, i) = 1 - (|\mathcal{S}| - 1)/N$ for $i \in [N]$ and $\mathbb{E}_{\mathcal{S}}[\mathbf{W}_{\mathcal{S}}](i, j) = (|\mathcal{S}| - 1)/N(N - 1)$ for $i \neq j$. Note that $\mathbb{E}_{\mathcal{S}}[\mathbf{W}_{\mathcal{S}}]$ has all positive entries. Therefore, we use the fact $\mathbf{T}^T \mathbf{T} = \mathbf{I}$ for permutation matrix $\mathbf{T}$ such that $\|\mathbb{E}[\mathbf{W}] - \mathbf{J}\| = \|\mathbb{E}_{\mathcal{S}, \mathcal{S}'}[\mathbf{W}_{\mathcal{S}}^T \mathbf{T}_{\mathcal{S}, \mathcal{S}'}^T \mathbf{T}_{\mathcal{S}, \mathcal{S}'} \mathbf{W}_{\mathcal{S}}] - \mathbf{J}\| = \|\mathbb{E}_{\mathcal{S}}[\mathbf{W}_{\mathcal{S}}^T \mathbf{W}_{\mathcal{S}}] - \mathbf{J}\| = \|\mathbb{E}_{\mathcal{S}}[\mathbf{W}_{\mathcal{S}}] - \mathbf{J}\| < 1$ by Perron–Frobenius theorem and eigendecomposition, which satisfies Assumption 2.5-ii).

## F.2 Discussion on partial client sampling

The commonly used partial client sampling algorithm in the FL literature [50, 30] is FedAvg as follows:

1. At time $n$, the central server updates its global parameter $\theta_n = \frac{1}{|\mathcal{S}|} \sum_{i \in \mathcal{S}} \theta_n^i$ from the agents in the previous set $\mathcal{S}$. Then, the central server selects a new subset of agents $\mathcal{S}'$ and broadcasts $\theta_n$ to agent $i \in \mathcal{S}'$, i.e., $\theta_n^i = \theta_n$;

2. Each selected agent $i$ computes $K$ steps of SGD locally and consecutively updates its local parameter $\theta_{n+1}^i, \cdots, \theta_{n+K}^i$ according to (1a);

3. Each selected agent $i \in \mathcal{S}'$ uploads $\theta_{n+K}^i$ to the central server.

Then, the central server repeats the above three steps with $\theta_{n+K}$ and a new set of selected agents.

In our client sampling scheme, at the aggregation step $n$, the design of $\mathbf{W}_{\mathcal{S}}$ results in $\tilde{\theta}_n^i = \frac{1}{|\mathcal{S}|} \sum_{j \in \mathcal{S}} \theta_n^j$ for a selected agent $i \in \mathcal{S}$, and $\tilde{\theta}_n^i = \theta_n^i$ for an unselected agent $i \notin \mathcal{S}$. Meanwhile, the central server updates the global parameter $\tilde{\theta}_n = \tilde{\theta}_n^i$ for $i \in \mathcal{S}$. Then, the permutation matrix $\mathbf{T}_{\mathcal{S} \to \mathcal{S}'}$ ensures that the newly selected agent $i \in \mathcal{S}'$ will use $\tilde{\theta}_n$ as the initial point for its subsequent SGD iterations. Consequently, from the selected agents' perspective, the communication matrix $\mathbf{W} = \mathbf{T}_{\mathcal{S} \to \mathcal{S}'} \mathbf{W}_{\mathcal{S}}$ corresponds to step 1 in FedAvg. As we can observe, both algorithms update the global parameter identically from the central server's viewpoint, rendering them mathematically equivalent regarding the global parameter update.

We acknowledge that under the *i.i.d* sampling strategy, the behavior of unselected agents in our algorithm differs from FedAvg. Specifically, unselected agents are idle in FedAvg, while they continue the SGD computation in our algorithm (despite not contributing to the global parameter update). Importantly, when an unselected agent is later selected, the central server overwrites its local parameter during the broadcasting process. This ensures that the activities of agents when they are unselected have no impact on the global parameter update.

To our knowledge, the FedAvg algorithm under the *Markovian* sampling strategy remains unexplored in the FL literature. Extrapolating the behavior of unselected agents in FedAvg from *i.i.d* sampling to Markovian sampling suggests that unselected agents would remain idle. In contrast, our algorithm enables unselected agents to continue evolving $X_n^i$. These additional transitions contribute to faster mixing of the Markov chain for each unselected agent and a smaller bias of $F_i(\theta, X_n^i)$ relative to its mean field $f_i(\theta)$, potentially accelerating the convergence.

## G  Additional Simulation

### G.1  Simulation Setup in Section 4

This simulation is performed on a PC with an AMD R9 5950X, RTX 3080 and 128 GB RAM. In this simulation, we assume that agents follow the DSGD algorithm (1). In Figure 2(a) - 2(c), each

---

[6]In Appendix F.2, we will discuss the mathematical equivalence between our client sampling strategy and the commonly used one in the FL literature [50, 30].

agent holds a disjoint local dataset (non-overlapping data points for every agent), while we distribute the ijcnn1 dataset [14] with more varied distribution among 100 agents by leveraging Dirichlet distribution with the default alpha value of 0.5 in Figure 2(d) - 2(f).

Moreover, we assume that all agents are distributed over a communication network. In order to create this network among 100 agents and the graph-like dataset structure held by each agent, we utilize connected sub-graphs from the real-world graph *Facebook* in SNAP [49]. All 100 agents collaborate together to generate a deterministic communication matrix $\mathbf{W} = [W_{ij}]$ with Metropolis Hasting algorithm of the following form: For $i, j \in [N]$, we have

$$W_{ij} = \begin{cases} \min\left\{\frac{1}{d_i}, \frac{1}{d_j}\right\} & \text{if agent } j \text{ is the neighbor of agent } i, \\ 0 & \text{otherwise,} \end{cases}$$

$$W_{ii} = 1 - \sum_{j \in [N]} W_{ij},$$

where $d_i$ represents the degree of agent $i$ in the graph. The communication interval $K$ is set to 1, as is the usual choice in DSGD [71, 80, 61, 68].

For the first group of agents, we assume they have full access to their datasets, thus performing *i.i.d.* sampling or single shuffling. In particular,

- *i.i.d.* sampling employed by agent $i$ means that the data point $X_n^i$ is independently and uniformly sampled from its dataset $\mathcal{X}_i$ at each time $n$.
- Single shuffling, by its name, only shuffles the dataset once and adheres to that specific order throughout the training process.

On the other hand, within the second group of agents, we assume that they hold graph-like datasets. Now, we introduce simple random walk (SRW), non-backtracking random walk (NBRW), and self-repellent random walk (SRRW) in order:

- SRW refers to the walker that chooses one of the neighboring nodes uniformly at random.
- NBRW, as studied in [2, 48, 5], is a variation of SRW, which selects one of the neighbors uniformly at random, with the exception of the one visited in the last step.
- SRRW, recently proposed by [25], is designed with a nonlinear transition kernel $\mathbf{K}[\mathbf{x}] \in [0, 1]^{N \times N}$ of the following form:

$$K_{ij}[\mathbf{x}] \triangleq \frac{P_{ij}(x_j/\mu_j)^{-\alpha}}{\sum_{k \in [N]} P_{ik}(x_k/\mu_k)^{-\alpha}}, \quad \forall i, j \in [N], \tag{89}$$

where matrix $\mathbf{P} = [P_{ij}]$ is the transition kernel of the baseline Markov chain and $\boldsymbol{\mu} = [\mu_i]$ is its corresponding stationary distribution. Additionally, $\alpha$ denotes the force of self repellence, and larger $\alpha$ leads to stronger force of self repellence, thus higher sampling efficiency [25, Corollary 4.3]. Moreover, vector $\mathbf{x} \in \mathbb{R}^N$ is in the interior of probability simplex, representing the empirical distribution, in other words, the visit frequency of each node in the graph. The update rule of this empirical distribution is in the following form:

$$\mathbf{x}_{n+1} = \mathbf{x}_n + \beta_{n+1}(\boldsymbol{\delta}_{X_{n+1}} - \mathbf{x}_n), \tag{90}$$

where $\beta_n \triangleq (n+1)^{-b}$ is the step size of SRRW iterates. $b = 1$ was original proposed in [25] and is recently extended to $b \in (0.5, 1)$ in [39]. In this simulation, we use SRW as the baseline Markov chain of SRRW, and in turn $\boldsymbol{\mu}$ is proportional to the degree distribution. We also assume $\mathbf{x}_0 = \mathbf{1}/N$, i.e., each node has been visited once, and choose the step size $\beta_n = (n+1)^{-0.8}$, force of self repellence $\alpha = 20$ according to the suggestion in [39, Section 4].

Since SRW, NBRW, and SRRW all admit the stationary distribution that is proportional to degree distribution, in order to obtain unfirom target in (15), we need to reweight the gradient computed by each agent $i$ in order to maintain an asymptotic unbiased gradient. Thus, agent $i$ should modify its SGD update from (1a) to the following:

$$\theta_{n+1/2}^i = \theta_n^i - \gamma_{n+1} \nabla F_i(\theta_n^i, X_n^i)/d(X_n^i). \tag{91}$$

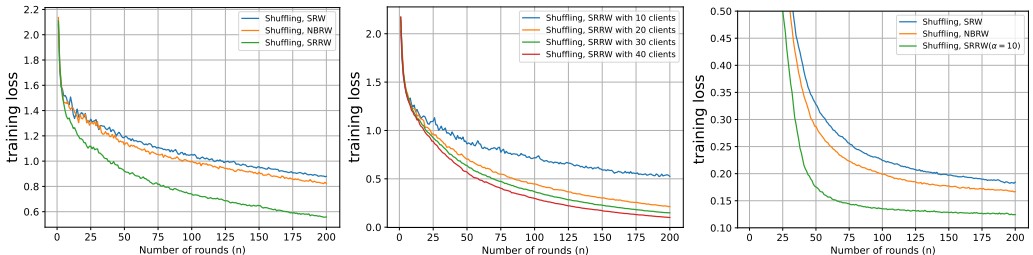

Figure 3: Image classification experiment. From left to right: (a) Comparison of various sampling strategies in image classification problem using 5-layer neural network. (b) Train a 5-layer CNN model with different number of total agents (clients) to show the linear speedup effect. (c) Train ResNet-18 model with different sampling strategies among 10 agents with participation ratio 0.4.

## G.2 Image Classification Task

In this part, we perform the image classification task through a 5-layer neural network, where the CIFAR10 dataset [44] with 50k image data is evenly distributed to 10 agents. Each agent possesses 5k images, which are further divided into 200 batches, each batch with 25 images.

The Convolutional Neural Network (CNN) model used in this simulation encompasses:

- Two convolutional layers (i.e., *nn.Conv2d(3, 6, 5)* and *nn.Conv2d(6, 16, 5)*), each followed by ReLU activation functions to introduce non-linearity and max pooling (i.e., *nn.MaxPool2d(2, 2)*) to reduce spatial dimensions.
- Three fully connected (linear) layers, concluding with a softmax output to handle the multi-class classification problem.

Similar to the simulation setup in Section 4, among the 10 participating agents, five have unrestricted access to their respective data allocations, enabling them to utilize the shuffling method to iterate through their batches. The other five agents are designed to simulate limited data access scenarios. Their data access is structured using five distinct graph topologies extracted from the SNAP dataset collection [49], each graph simulating a unique communication pattern among the batches (nodes) of data. Within these topologies, agents adopt one of three random walk strategies — SRW, NBRW, and SRRW, all with importance reweighting — to sample the batches for training.

Local model training is conducted for five epochs at each agent before aggregating the updates at a central server — a process repeated for a total of 200 communication rounds. Each epoch consists of a full traversal of the local dataset of agents in the first group, or 200 batches sampled for training in the second group of agents. To mimic realistic conditions, we also introduce partial agent participation where only $40\%$ of agents are selected randomly to transmit their updates in each round, reflecting the intermittent communication in real-world FL deployments. Lastly, the selection of the step size $\beta_n$ for SRRW iterates (90) is a critical aspect of our experiments. In this simulation, we experiment with various values of $b \in \{0.501, 0.6, 0.7, 0.8, 0.9\}$ to determine the most advantageous setting for maximizing the benefits of the SRRW strategy. Based on our findings, the best choice for the SRRW step size is $b = 0.501$, in other words, $\beta_n = (n+1)^{-0.501}$.

The simulation result is quantified by averaging the training loss across ten repeated trials for each configuration. As depicted in Figure 3(a), the training results are consistent with our previous findings in Figure 2(a) in the context of the FL framework and the training of neural networks: the use of a more effective sampling approach, even for a portion of the agents, results in significant enhancements in the overall training of the model, and this improvement is further enhanced through the highly efficient sampling strategy SRRW.

In Figure 3(b) and 3(c), we perform image classification experiments in the FL setting with partial client participation. Only 4 random agents will participate in the training process at each aggregation phase. In Figure 3(b), we fix the sampling strategy (shuffling, SRRW with $\alpha = 10$) and test the linear speedup effect for the 5-layer CNN model by duplicating 10 agents to $N$ agents with $N \in \{10, 20, 30, 40\}$, keeping the same participation ratio $0.4$. As can be seen from the plot, the training loss is inversely proportional to the number of agents, i.e., at 200 rounds, the training loss is $0.52$ for 10 agents, $0.23$ for 20 agents, $0.18$ for 30 agents, and $0.12$ for 40 agents. In Figure 3(c), we extend the current simulation from 5-layer CNN model to ResNet-18 model [34] in order to

numerically test the performance of different sampling strategies in a more complex neural network training task. By fixing the shuffling in the first group of agents, we observe that improving Markovian sampling from SRW to NBRW, then to SRRW, gives accelerated training process with smaller training loss.

# H Limitations

Our study provides crucial insights into the identification of nuanced agents' sampling behaviors in UD-SGD, where improving each agent's sampling strategy speeds up overall system performance without additional computational burden except the additional storage for the visit counts used for sampling their datasets. Our UD-SGD is scalable in terms of larger datasets as the sampling strategy (i.e., random walk) utilized by each agent leverages only local information for its dataset. However, our work has two limitations that should be acknowledged.

1. Assumption 2.4 posits that the parameter trajectory $\{\theta_n\}$ is almost surely bounded, which is a strong assumption. This is crucial for guaranteeing the well-defined nature of all related quantities. Some mechanisms such as projections onto a compact subset [45, Chapter 5.1], or truncation-based method with expanding compact subsets can do the trick to ensure that the iteration is always bounded [3]. As mentioned in our discussion after Assumption 2.4, only recently the stability of SGD under Markovian sampling has been guaranteed to hold for some class of objective function $f$ [9], while the discussion on stability issue under multi-agent scenario with Markovian sampling remains an open problem and we do not pursue to remove this stability assumption in this paper.

2. Asymptotic analysis: The main results of our work, i.e., almost sure convergence and central limit theorem in distributed optimization, are based on asymptotic analysis and might not accurately represent the finite-sample performance of each contributing agent in the system. The state-of-the-art finite-sample analysis in the literature only focuses on the worst-performing agent that cannot capture the nuanced differences between agents' sampling strategies, with the explanation detailed in Footnote 2. Thus, a finite-sample error bound that distinguish every agent's dynamics is still unknown and regarded as a future direction.

