# OpenReview forum: "Does Worst-Performing Agent Lead the Pack? Analyzing Agent Dynamics in Unified Distributed SGD"
_NeurIPS.cc/2024/Conference — NeurIPS 2024 poster_

### Official Review · Reviewer_ds2S · 2024-07-09

**Soundness:** 3
**Presentation:** 3
**Contribution:** 3
**Rating:** 7
**Confidence:** 3

**Summary:**

This paper provides an asymptotic analysis of Unified Distributed SGD (UD-SGD) under heterogeneous agent dynamics and a large family of communication topology. It shows that under certain assumptions: i) regularity of the gradient, **ii) Ergodicity of Markovian sampling**, iii) decreasing step size and intervals not increasing so fast, iv) stability on model parameter v) contraction property of douly-stochastic communication matrix, UD-SGD algorithm guarantees that every agent parameter $\theta_n$ converges to the same minimum value $\theta^*$ with some variance matrix $V$ and the mean value of all agents $\overline{\theta}=\sum_{i=1}^n\theta_i$ also converges to $\theta_i$ with some variance matrix $V'$.

**Strengths:**

1. Although some previous works discuss the asymptotic analysis for distributed learning, most of them focus on some specific algorithm under a given communication topology and i.i.d sampling. This work is more general. It discusses the asymptotic analysis for more diverse communication topologies with the Markov sampling method.
2. In order to discuss the theoretical properties of a general distributed learning framework under diverse communication topologies, the author introduce the Unified Distributed SGD (UD-SGD). The clear definition for a unified version of DSGD helps readers better understand the universality of this property.
3. Let $V_i$ be the limiting covariance matrix of agent i and $V$ be the covariance matrix of the mean of all agents ($V=\frac{1}{n^2}\sum_{i=1}^{n} V_i$). The author provides the exact for of $V$.
4. The provides numerical experiments on logistic regression and neural networks with different sampling strategies to support their theory part.

**Weaknesses:**

1. In line 26, the authors claim that $\mathcal{L}$ represents the collection of local minima of objective function $f$. In line 246, the authors claim that $\theta^*\in \mathcal{L}$. Since when $f$ is non-convex, $\mathcal{L}$ do not have a single element and $\theta^*$ could have a lot of selections. How could we ensure that the convergence of consensus $\theta_n$ is unique? And how to choose $\theta^*$? Will that be the global minima?
2. The asymptotic analysis Theorem 3.3 is mainly based on Assumption 2.2, which assumes that the dynamic agent $\{X_{n}^i\}$ is an ergodic Markov chain. Could the author give some sampling strategies that satisfy this property and are now widely used in DSGD? What are the advantages of Markov sampling compared with the i.i.d sampling method?
3. In figure 2 of the experimental section, the authors only compare the performance of different Markov sampling methods. Could the author also compare with the iid sampling and sampling shuffle strategies?

**Questions:**

Same as the weakness.

**Limitations:**

Same as the weakness.

---

> ### Author Rebuttal · Authors · 2024-08-06
>
> > Q1: In line 26, the authors claim that $\\mathcal{L}$ represents the collection of local minima of objective function $f$. In line 246, the authors claim that $\\theta^*\\in\\mathcal{L}$. Since when $f$ is non-convex, $\\mathcal{L}$ do not have a single element and $\\theta^*$ could have a lot of selections. How could we ensure that the convergence of consensus $\\theta_n$ is unique? And how to choose $\\theta^*$? Will that be the global minima?
>
> As mentioned in Footnote 1 of our current manuscript, our non-convex setting only concerns the (possibly many) local minima, which is a common scenario in deep learning. We believe that the reviewer’s confusion comes from our wording ‘an optimal point $\\theta^*\\in\\mathcal{L}$’ around line 246. Here $\\theta^*$ refers to a local minima rather than a global minima. In our UD-SGD framework, we ensure that all agents reach a consensus and converge to one of the local minima. Each run of the algorithm may converge to different local minima depending on the initialization and the stochastic nature of the updates. To avoid confusion, we will revise the wording from “an optimal point” to “a local minimum” in line 246. This should make it clear that we are discussing convergence to a local minima rather than a global minima.
>
> Our CLT result is conditional on the algorithm converging to a specific local minima $\\theta^*$. Given that $\\lim_{n\\to\\infty} \\theta\_n=\\theta^*$, we derive the CLT for this particular $\\theta^*$. Without knowing the exact local minima to which the algorithm converges, it is impossible to compute the limiting covariance, which indicates the performance at the converging point. We appreciate the reviewer’s feedback and will add the following clarification to line 252 in the revision: ‘For notational simplicity, and without loss of generality, our remaining results are stated while conditioning on the event that $\\{\\theta\_n\\to\\theta^*\\}$ for some $\\theta^*\\in\\mathcal{L}$’.
>
> Lastly, we mentioned in Footnote 1 that additional condition like the PL inequality is required to guarantee convergence to a global minima because the PL inequality ensures that every local minima is also a global minima. This simplifies the landscape of the objective function. However, our current work focuses on local minima and does not pursue the PL inequality.
>
> > Q2: The asymptotic analysis Theorem 3.3 is mainly based on Assumption 2.2, which assumes that the dynamic agent is an ergodic Markov chain. Could the author give some sampling strategies that satisfy this property and are now widely used in DSGD? What are the advantages of Markov sampling compared with the i.i.d sampling method?
>
> Metropolis-Hasting Random Walk (MHRW) and Simple Random Walk (SRW) are the most popular sampling strategies in DSGD, which satisfy the ergodic Markov chain property (e.g., Section 2.1 in [1], Section 2 in [2]). Specifically, MHRW with uniform target distribution has the following transition probabilities:
> $$P(X\_{n+1}=j | X\_n=i) \\triangleq P\_{ij} = \\min \\{\\frac{1}{d\_i},\\frac{1}{d\_j}\\} ~~\forall i \\neq j, P\_{ii}= 1-\\sum\_{j\\neq i} P\_{ij}$$
> For SRW, the transition probability is $P\_{ij}=1/d\_i$ for $j\\in Neighbor(i)$ and $P\_{ii}=0$, with the stationary distribution proportional to the degree of each node.
>
> Regarding the advantages of Markovian sampling over iid sampling, we have emphasized in lines 61 - 72. For the sake of completeness, here is a brief summary:
>
> - **Handling Limited Data Access:** Markovian sampling is particularly useful when agents have limited or sequential access to data, making i.i.d. sampling infeasible. For example, Fog learning considers a multi-layer hybrid learning framework consisting of heterogeneous devices, where each agent is treated as an edge server of the next-layer network [3]. This contributes to the graph-like structure of the dataset held by each agent, where i.i.d. sampling is infeasible.
>
> - **Efficiency in High-Dimensional Spaces:** In high-dimensional data spaces (such as constraints), Markov Chain Monte Carlo (MCMC) methods, such as those employing Markovian sampling, are more computationally efficient and effective compared to i.i.d. sampling, which can be costly and less practical because multiple rejections can happen before obtaining a sample generated by i.i.d. sampling that satisfies constraints.
>
>
> > Q3: In figure 2 of the experimental section, the authors only compare the performance of different Markov sampling methods. Could the author also compare with the iid sampling and sampling shuffle strategies?
>
> Thank you for the suggestion. In the current manuscript, we have included a comparison between i.i.d. sampling and shuffle methods, which was mentioned in lines 357 – 359 with details deferred to Figure 3 in Appendix G.1 due to space constraints. In particular, we fixed the second group of clients who perform Markovian sampling and varied the sampling strategies for the first group of clients. The results indicate that the shuffle method outperforms i.i.d. sampling in terms of MSE. This finding is consistent with existing literature (e.g., [2]), which shows that the shuffle method has zero asymptotic variance, in contrast to the positive variance associated with i.i.d. sampling. We will put this comparison from Appendix back to Section 4 if our paper is accepted (with one extra page for the camera-ready version).
>
> [1] B. Johansson, M. Rabi, and M. Johansson. "A randomized incremental subgradient method for distributed optimization in networked systems." SIAM Journal on Optimization 20, no. 3 (2010): 1157-1170.
>
> [2] J. Hu, V. Doshi, and D.Y. Eun. "Efficiency ordering of stochastic gradient descent." NeurIPS, 2022.
>
> [3] S. Hosseinalipour, C.G. Brinton, V. Aggarwal, H. Dai, and M. Chiang. "From federated to fog learning: Distributed machine learning over heterogeneous wireless networks." IEEE Communications Magazine 58, no. 12 (2020): 41-47.

---

> > ### Comment · Reviewer_ds2S · 2024-08-11
> >
> > Thanks to the authors for their detailed response. They answer all my concerns and I would like to keep my score.

---

### Official Review · Reviewer_WSiF · 2024-07-09

**Soundness:** 3
**Presentation:** 3
**Contribution:** 3
**Rating:** 6
**Confidence:** 3

**Summary:**

This paper studies the asymptotic convergence behavior of federated learning under the UD-SGD framework with Markovian data. The authors establish a new central limit theorem that considers the strategy of every agent, which goes beyond the existing bounds that only focus on the worst-performing agent. Their theory emphasizes the importance of every individual and also explains linear speedup and asymptotic network independence.

**Strengths:**

1. This paper established theories for a more general federated learning framework and gives a refined analysis to encode the behavior of every agent into their bound.
2. The new asymptotic analysis could explain linear speedup scaling with the number of agents and asymptotic network independence.
3.  The new analysis shows that upgrading only a small group of agents could benefit the whole system and the authors conduct experiments to validate it.
4. The paper is well-written.

**Weaknesses:**

1. As the author mentioned, Assumption 2.4 seems strong and finite-sample analysis is preferred,
2. As far as I could see in the experiments, the author only implemented DSGD but the UD-SGD framework has more algorithms.
3. The linear speedup scaling with the number of agents and asymptotic network independence are only mentioned in theory but do not have empirical justification.
4. The experiments for neural networks are somewhat toy.

**Questions:**

1. What is the main technical challenge when utilizing Poisson equation to prove Theorem 3.3? Specifically, what is new compared with the analysis in reference [23] and [30]?
2. Do the authors believe the current asymptotic analysis technique could be extended to (federated) reinforcement learning? I saw some finite-sample analyses for federated RL in your references like [72].

**Limitations:**

Yes.

---

> ### Author Rebuttal · Authors · 2024-08-06
>
> > Q1: What is the main technical challenge when utilizing Poisson equation to prove Theorem 3.3? Specifically, what is new compared with the analysis in reference [23] and [30]?
>
> The main technical challenge in utilizing the Poisson equation to prove Theorem 3.3 lies in addressing the consensus error in the decomposed noise terms for each agent. Specifically:
>
> ---
> *We decompose $\\nabla F\_i (\\theta\_n^i,X\_n^i )-\\nabla f\_i (\\theta\_n^i)$ into three parts in (48) using Poisson equation: $e\_{n+1}^i,\\nu\_{n+1}^i,\\xi\_{n+1}^i$. The consensus error $\\theta\_n^i-\\theta\_n$ embedded in noise terms $e\_{n+1}^i$ and $\\xi\_{n+1}^i$ is a new factor, whose characteristics have been quantified in our Lemma 3.1 but are not present in the single-agent scenario analyzed as an application of stochastic approximation in references [23] and [30].*
>
> ---
> We will replace the explanation of technical challenges around lines 259-261 in our original submission with above contents. Full explanations are provided below for the completeness. All line numbers and equation numbers are referred to our original submission.
>
> As mentioned in lines 754 – 756, for the noise term $\\xi\_{n+1}^i$, directly following the analysis in [23] and [30] leads to a one-step error for all agents, as expressed in (53). In a single-agent scenario (like SGD), this one-step error is simply $-\gamma_n\\nabla F$. However, in a multi-agent scenario, this one-step error cannot be directly analyzed due to the model parameters $\\Theta\_n$ of all agents being multiplied by the communication matrix $\\mathbf{W}$ in (18). To overcome this issue, we decompose this one-step error among all agents into two parts in (54): the consensus error among all agents and the one-step error of the average solution among all agents. Although separating the error into consensus error and error between the average and target solution has been done similarly in [1], our approach in quantifying the consensus error in Lemma 3.1 handles broader classes of communication patterns within our UD-SGD framework, instead of DSGD in optimization in [1]. We can separate the effect of the consensus error and utilize our Lemma 3.1 on the speed of consensus error to obtain the condition on $\\xi\_{n+1}^i$ in (55). Similarly, for the noise term $e\_{n+1}^i$, we follow a similar logic in decomposing its covariance in (73). This separation of the consensus error allows us to derive (75), ensuring that we can rigorously analyze the covariance structure and account for the multi-agent interactions.
>
> [1] Zeng, Sihan, Thinh T. Doan, and Justin Romberg. "Finite-time convergence rates of decentralized stochastic approximation with applications in multi-agent and multi-task learning." IEEE Transactions on Automatic Control 68, no. 5 (2022): 2758-2773.
>
> > Q2: Do the authors believe the current asymptotic analysis technique could be extended to (federated) reinforcement learning? I saw some finite-sample analyses for federated RL in your references like [72].
>
> Yes, we believe that our asymptotic analysis can be extended to Federated RL. Our analysis is based on transforming the UD-SGD framework into a stochastic approximation setting, which includes both optimization algorithms (SGD) and RL algorithms (TD learning and Q-learning). This structural compatibility allows for an extension to RL.
>
> To extend our asymptotic analysis to Federated RL, we need to make the following adaptations: Replace the gradient $-\\nabla F\_i$ in our UD-SGD framework with the TD error from the Bellman equation, denoted as $g\_i (\\cdot)$ in Algorithm 1 of [72]; Adjust the CLT in Theorem 3.3 by replacing $-\\nabla^2 F\_i$ with $\\nabla g\_i$ to reflect the gradient of the TD error. This adaptation should yield the statistical properties of Federated RL, providing insights into the asymptotic performance of each agent rather than focusing solely on the worst-performing agent, as done in some finite-sample analyses like [72].
>
> A notable caveat in RL is that Markovian samples are derived from a given behavioral policy, which is often uncontrollable. While our analysis can still evaluate the contribution of each agent to the overall system performance, the improvements from efficient sampling strategies, as discussed in our paper, cannot be directly extrapolated from optimization setting to the RL context.
>
> > Other concerns: As far as I could see in the experiments, the author only implemented DSGD but the UD-SGD framework has more algorithms. The linear speedup scaling with the number of agents and asymptotic network independence are only mentioned in theory but do not have empirical justification. The experiments for neural networks are somewhat toy.
>
> We appreciate the reviewer's request for more empirical results to support our theory. To address your concerns, we conducted additional experiments shown in Figures 1–3 of the rebuttal:
>
> - In Figure 1(c), we simulate DSGD with a randomly chosen communication matrix from 5 doubly stochastic matrices at each aggregation phase. In Figure 1(d), we test the DFL algorithm with increasing communication interval $K(l) = \\max\\{1,⌈log⁡(l)⌉\\}$ after $l$-th aggregation phase. Moreover, we trained a ResNet-18 model with different sampling strategies, as shown in Figure 2(b). These results demonstrate that improved sampling strategies lead to faster convergence with smaller MSE.
>
> - To numerically test the linear speedup, We conducted image classification on a 5-layer CNN for the CIFAR-10 dataset with partial client participation. We varied the number of agents from 10 to 40. As shown in Figure 2(a), training loss inversely proportional to the number of agents confirms the linear speedup.
>
> - In Figure 1(b), we observe that all four algorithms (Centralized SGD, DSGD with time-varying topologies, FL with full client participation, and DFL with increasing communication interval) reach similar performance around 1000 steps, indicating asymptotic network independence.

---

> > ### Comment · Reviewer_WSiF · 2024-08-10
> >
> > Thank you for the detailed response. I do not have further questions now.

---

### Official Review · Reviewer_Evbf · 2024-07-18

**Soundness:** 4
**Presentation:** 3
**Contribution:** 4
**Rating:** 7
**Confidence:** 4

**Summary:**

This paper conducts an asymptotic analysis of Unified Distributed SGD (UD-SGD), which has a generalized communication patterns (modelled with a doubly stochastic communication matrix). The paper investigates several different sampling strategies, such as i.i.d. sampling, shuffling, and Markovian sampling

**Strengths:**

- Rigorous theory: I went through most of the proofs in the appendix and as far as I can tell the theory is sound.
- Well presented: Despite the heavy derivation I find this paper easy to follow as the proof strategies and theoretical insights are discussed clearly in the main exposition.

**Weaknesses:**

Empirical results are quite sparse. Not that this is a big problem for a theory paper, but maybe some additional studies would make it more interesting. I have some suggestions:
- Are all of the results obtained using the same communication matrix (referring to line 982 in appendix G1)? Since the theory is generic for all double stochastic W maybe the authors could present the average result over 5 random W?
- How was local data allocated to each client? The text said each client holds 500 data points from a bigger dataset -- are they iid? If so, it might be interesting to investigate non iid local data (using Dirichlet distribution, which is a common setting in FL)
- How does client dropout (also a common setting in FL) affect the theory and empirical results?

**Questions:**

I have no further question.

**Limitations:**

No negative societal impact.

---

> ### Author Rebuttal · Authors · 2024-08-06
>
> > Q1: Are all of the results obtained using the same communication matrix (referring to line 982 in appendix G1)? Since the theory is generic for all double stochastic W maybe the authors could present the average result over 5 random W?
>
> We thank the reviewer for the question and the suggestion. To clarify, we didn’t use the same communication matrix throughout all simulation results in the current manuscript. For the simulation in Section 4, we indeed used a fixed doubly stochastic matrix $\\mathbf{W}$ (with expression in line 982), as pointed out by the reviewer.
>
> In the simulation in Appendix G.2, we have conducted the experiment in the FL setting with partial client participation. In this scenario, at each aggregation phase, only 4 random agents (clients) out of 10 upload their model parameters to a central server. This results in a random communication matrix $\\mathbf{W}\_{\\mathcal{S}}$, determined by a random set of participating clients $\\mathcal{S}$ at each aggregation phase. The detailed expression of this random matrix has been provided in Appendix F.1. We admit that the random nature of the matrix $\\mathbf{W}\_{\\mathcal{S}}$ was not explicitly explained in the simulation setup in Appendix G.2. To improve clarity, we will revise the description around lines 374-376 in the main body and explicitly point out in Appendix G.2 that the communication matrix in the FL setting with partial client participation is stochastic.
>
> To accommodate the reviewer’s suggestions, we also simulate 5 random W setting under the decentralized SGD scenario (namely DSGD with time-varying topologies (DSGD-VT)), where at each aggregation phase, we randomly pick a communication matrix $\\mathbf{W}$ from a set of 5 doubly stochastic matrices. We perform the same configuration of sampling strategies as in Section 4, where the first group of agents perform shuffling or i.i.d. sampling while the rest of agents conduct simple random walk (SRW), non-backtracking random walk (NBRW), and self-repellent random walk (SRRW). The result is shown in Figure 1(c) in the rebuttal pdf file (could be found in author rebuttal). We observe the same numerical trend as other algorithms in the current manuscript (DSGD with fixed $\\mathbf{W}$ in Section 4, FL with partial client participation in Appendix G.2), where improving agent’s sampling strategies leads to faster convergence with smaller MSE.
>
> > Q2: How was local data allocated to each client? The text said each client holds 500 data points from a bigger dataset -- are they iid? If so, it might be interesting to investigate non iid local data (using Dirichlet distribution, which is a common setting in FL)
>
> Thanks for raising the question about IID and Non-IID data distribution. First, our theory is applicable to both IID and Non-IID data settings. Regarding the simulation in Section 4, as pointed out in lines 340 – 341 in the current manuscript, the ijcnn1 dataset (50k data points with binary classes) was shuffled and then evenly split among 100 agents, with each agent receiving 500 data points. Thus, each agent held a disjoint local dataset such that they didn’t have overlapped data point with other agents, resulting in Non-IID data. In our original simulation, the distribution of labels was relatively balanced, with the number of label '1' for each agent ranging from 40 to 60.
>
> We agree that investigating more diverse data distributions would provide valuable insights, especially in federated learning. To address this, we conduct additional simulations in our rebuttal, where the data is allocated to agents using a Dirichlet distribution with the default alpha value of 0.5, which ensures a more varied distribution of labels among agents. For the binary classification problem, Figure 1(a) in the rebuttal pdf file indicates that the number of data with label '1' ranges from 0 to 350 across different agents, forming the Non-IID data. We simulate DSG and DFL based on this imbalanced dataset and the results are shown in Figure 1 (c) & (d). The plots demonstrate that our asymptotic analysis is invariant to data distribution with the same trend for different combinations of sampling strategies.
>
> > Q3: How does client dropout (also a common setting in FL) affect the theory and empirical results?
>
> Thank the reviewer for pointing out the dropout phenomenon. In terms of client dropout, we believe that the reviewer refers to the following behavior:
>
> *The client temporarily exits the training process and will rejoin the training in the future with some probability (Bernoulli dropout).*
>
> This is usually modeled as ‘partial client participation’ in FL. In our original submission, we have discussed this scenario in both theory (in Appendices F.1 and F.2) and in experiment (only 4 random participants out of 10 agents at each aggregation in Appendix G.2). Even under this client dropout effect, the random communication matrix $\\mathbf{W}$ generated by this ‘client dropout’ effect still satisfies our assumption 2.5. Therefore, our UD-SGD framework guarantees the convergence to the set of local minima because every agent equitably contributes to the learning process in the long run. In Figure 4 of our original submission, as well as Figure 2(b) in the rebuttal pdf file, we have shown that in the FL with partial client participation, our main message in this paper still holds: efficient sampling strategies employed by individual agents improve the overall convergence in UD-SGD.

---

> > ### Comment · Reviewer_Evbf · 2024-08-13
> > **Thank you for the detailed response**
> >
> > I don't find many weaknesses to begin with, and I'm happy to maintain my positive score.

---

### Author Rebuttal · Authors · 2024-08-06

We thank all three reviewers for their comments and for the time and effort they put into reading, understanding and evaluating our paper. In particular, we appreciate the question to make our technical contribution much clearer (Reviewer WSiF), and we should conduct more experiments to support our theory (Reviewer Evbf, WSiF, ds2S). We carefully consider all questions, concerns and comments provided by reviewers and address all of them appropriately. We provide detailed responses to each review separately and we believe that our responses address all of the reviewers' concerns. We also upload a PDF file containing figures of additional simulation results, which will be included in our revision. When we respond to each reviewer, we specify whether the figure comes from our original submission or from our additional experiments in the rebuttal.

Now, we explain our additional experiments in the PDF file. Fig. 1 shows the extra simulation settings in the binary classification problem with Non-IID data, and Fig. 2 includes more experiments on the image classification problem. Specifically,

---
1. We perform additional binary classification simulations on the IJCNN1 dataset with more varied distribution among 100 agents by leveraging Dirichlet distribution with the default alpha value of 0.5 (see Fig. 1(a) for the data distribution). We follow the same setup as in Section 4 of our original submission, i.e., we split 100 agents into two groups, where the first group of 50 agents leverages either iid sampling or shuffling, while the second group utilizes three Markovian sampling methods: SRW, NBRW, and SRRW (with hyperparameter $\\alpha=20$). All 100 agents exchange model parameters through a communication matrix $\\mathbf{W}$.

    In Fig. 1(b), we empirically show the asymptotic network independence property via four algorithms under our UD-SGD framework:
    - Centralized SGD (communication interval $K=1$, communication matrix $\\mathbf{W}=\\mathbf{1}\\mathbf{1}^T/N$);
    - LSGD-FP (FL with full client participation, $K=5$, $\\mathbf{W}=\\mathbf{1}\\mathbf{1}^T/N$);
    - DSGD-VT (DSGD with time-varying topologies, $K = 1, \\mathbf{W}$ randomly chosen from $5$ doubly stochastic matrices);
    - DFL (decentralized FL with fixed $\\mathbf{W}$ and increasing communication interval $K(l)=\\max\\{1,log(l)\\}$ after $l$-th aggregation).

    *We fix the sampling strategy (shuffling, SRRW) throughout this plot.* All four algorithms overlap around 1000 steps, implying that they have entered the asymptotic regime with similar performance where the CLT result dominates, implying the asymptotic network independence in the long run.

    Fig. 1(c) & (d) show the performance of different sampling strategies in the DSGD-VT and DFL algorithms in terms of MSE (over 120 independent trials). Both plots consistently demonstrate that improving agent’s sampling strategies (e.g., shuffling > iid sampling, and SRRW > NBRW > SRW) leads to faster convergence with smaller MSE, supporting our theory.
---
2. We perform additional image classification experiments on the CIFAR-10 dataset in the FL setting with partial client participation. We follow the same setup as in Appendix G.2 of our original submission, i.e., 50k image data are evenly distributed to 10 agents, 5 agents in the first group (iid sampling or shuffling) and 5 in the second group (Markovian sampling). Each agent possesses 5k *disjoint* images, which are further divided into 200 batches, each batch with 25 images. Only 4 random agents will participate in the training process at each aggregation phase. In Fig. 2(a), *we fix the sampling strategy (shuffling, SRRW with $\\alpha=10$)* and test the linear speedup effect for the 5-layer CNN model (structure given in lines 1028 - 1033 in our original submission) by duplicating 10 agents to $N$ agents with $N\\in\\{10,20,30,40\\}$, keeping the same participation ratio 0.4. As can be seen from the plot, the training loss is inversely proportional to the number of agents, i.e., at 200 rounds, the training loss is 0.52 for 10 agents, 0.23 for 20 agents, 0.18 for 30 agents, and 0.12 for 40 agents. In Fig. 2(b), we extend the current simulation in Appendix G.2 from 5-layer CNN model to ResNet-18 model in order to numerically test the performance of different sampling strategies in a more complex neural network training task. By fixing the shuffling in the first group of agents, we observe that improving Markovian sampling from SRW to NBRW, then to SRRW, gives accelerated training process with smaller training loss.

We eagerly anticipate feedback from the reviewers and are happy to offer further details and clarifications as needed!

---

### Decision · Program_Chairs · 2024-09-25

**Decision:**

Accept (poster)

**Comment:**

All reviewers agree that this is a good paper. It introduces theoretical advancements in a generalized distributed learning framework, offering a refined analysis that captures the behavior of individual agents. The new asymptotic analysis effectively explains linear speedup with the number of agents and demonstrates asymptotic network independence. The findings also show that upgrading even a small subset of agents can positively impact the entire system, a result validated by the experiments. Consequently, the paper is recommended for acceptance.